# Elevator-like movements of prestin mediate outer hair cell electromotility

Makoto F. Kuwabara ●[1,6], Bassam G. Haddad ●[2,6], Dominik Lenz-Schwab ●[1,6], Julia Hartmann[1,6], Piersilvio Longo ●[2], Britt-Marie Huckschlag[1], Anneke Fuß[1], Annalisa Questino[1], Thomas K. Berger ●[1], Jan-Philipp Machtens ●[2,3] ✉ & Dominik Oliver ●[1,4,5] ✉

The outstanding acuity of the mammalian ear relies on cochlear amplification, an active mechanism based on the electromotility (eM) of outer hair cells. eM is a piezoelectric mechanism generated by little-understood, voltage-induced conformational changes of the anion transporter homolog prestin (SLC26A5). We used a combination of molecular dynamics (MD) simulations and biophysical approaches to identify the structural dynamics of prestin that mediate eM. MD simulations showed that prestin samples a vast conformational landscape with expanded (ES) and compact (CS) states beyond previously reported prestin structures. Transition from CS to ES is dominated by the translational-rotational movement of prestin's transport domain, akin to elevator-type substrate translocation by related solute carriers. Reversible transition between CS and ES states was supported experimentally by cysteine accessibility scanning, cysteine cross-linking between transport and scaffold domains, and voltage-clamp fluorometry (VCF). Our data demonstrate that prestin's piezoelectric dynamics recapitulate essential steps of a structurally conserved ion transport cycle.

Solute carrier (SLC) 26 anion transporters are a family of versatile transport proteins that act either as coupled anion antiporters or as uncoupled, channel-like transporters of small anions such as chloride, iodide, and sulfate. Mammalian SLC26A5, known as prestin[1], however, stands out in lacking substantial anion transport capacity[2,3]. Instead, prestin confers a unique property to auditory sensory outer hair cells (OHCs), termed electromotility (eM), by which the cells rapidly and reversibly contract in response to depolarization of the membrane potential[4–6]. eM is thought to be the causative mechanism of cochlear amplification, an active mechanical process that is required for the excellent sensitivity and frequency selectivity of the mammalian ear[7–9].

eM is a piezoelectric process, where mechanical activity is directly driven by voltage in prestin-containing plasma membrane (reviewed in refs. 10–13). At the molecular level, prestin is thought to act by an area-motor mechanism, changing its circumference in the membrane in response to membrane potential variation. Aggregate area changes in the membrane plane of densely packed prestin molecules enforce cellular movement of OHCs[14–17]. Such a mechanism implies a major molecular rearrangement dependent on membrane potential. Despite attracting much interest since the discovery of prestin[1], this mechanism remained unsolved so far.

Several lines of indirect evidence suggest that prestin's mechano-electric activity is associated with the conformational changes that

[1]Department of Neurophysiology, Institute of Physiology and Pathophysiology, Philipps University Marburg, 35037 Marburg, Germany. [2]Institute of Biological Information Processing (IBI-1), Molekular- und Zellphysiologie, and JARA-HPC, Forschungszentrum Jülich, Jülich, Germany. [3]Institute of Clinical Pharmacology, RWTH Aachen University, Aachen, Germany. [4]DFG Research Training Group, Membrane Plasticity in Tissue Development and Remodeling, GRK 2213, Philipps University, Marburg, Germany. [5]Center for Mind, Brain and Behavior (CMBB), Universities of Marburg and Giessen, Marburg, Germany. [6]These authors contributed equally: Makoto F. Kuwabara, Bassam G. Haddad, Dominik Lenz-Schwab, Julia Hartmann. ✉e-mail: j.machtens@fz-juelich.de; oliverd@staff.uni-marburg.de

mediate anion transport in other members of the SLC26 family. Thus, non-mammalian prestin orthologs, although highly conserved in protein sequence, are electrogenic anion exchangers[2,3]. Further, the electromechanical activity of prestin requires binding of monovalent anions, usually chloride[18], into the same central anion-binding site that serves ion transport[19,20]. However, the molecular mechanism of anion transport in SLC26 proteins is still unclear. Recent atomistic structures of bacterial[21,22], plant[23], and mammalian SLC26 transporters[24,25] revealed the common molecular structure. Briefly, SLC26 proteins are obligate dimers[25,26]. The transmembrane domain of each protomer follows a 7 + 7 inverted repeat architecture with the transmembrane (TM) segments organized into two major helix bundles termed transport (previously, core) and scaffold (gate) domains[27] (Fig. 1a; Supplementary Fig. 1a). This overall architecture is shared by the SLC4 and SLC23 transporter families[27]. Comparison of various experimental structures suggests that transport in these SLC groups may be mediated by an elevator mechanism, i.e., a translational-rotational rigid-body movement of the entire transport domain along a largely static scaffold domain, affording alternating access of the central binding site[27–30].

Only recently, cryo-EM structures of mammalian prestin were reported[19,31,32] that are consistent with an earlier structural model[20] and exhibit all structural hallmarks of the SLC26 family. Strikingly, distinct conformations of prestin were resolved in the presence of different ligand anions. Under physiological conditions with chloride present, structures with the central anion-binding site in an intermediate, or occluded, state were obtained[19,31,32]. Distinct conformations were seen when chloride was substituted by salicylate or sulfate, which compete with chloride and thereby inhibit eM[2,18]. With sulfate, inward-facing states were resolved[19,31], similar to previously reported conformations of other SLC26 transporters. Comparison of 'Up' and 'Down' states (obtained with Cl⁻ and sulfate, respectively) suggested an elevator-like translation of the transport domain. Notably, the cross-sectional area of the prestin dimer substantially differed between conformations, with the 'Up' state being more compact[19,31]. The different calibers suggested that the observed states may outline the molecular transitions underlying prestin's area-motor activity.

However, to account for eM, transitions between states need to be voltage-dependent, which cannot be shown by cryo-EM, where

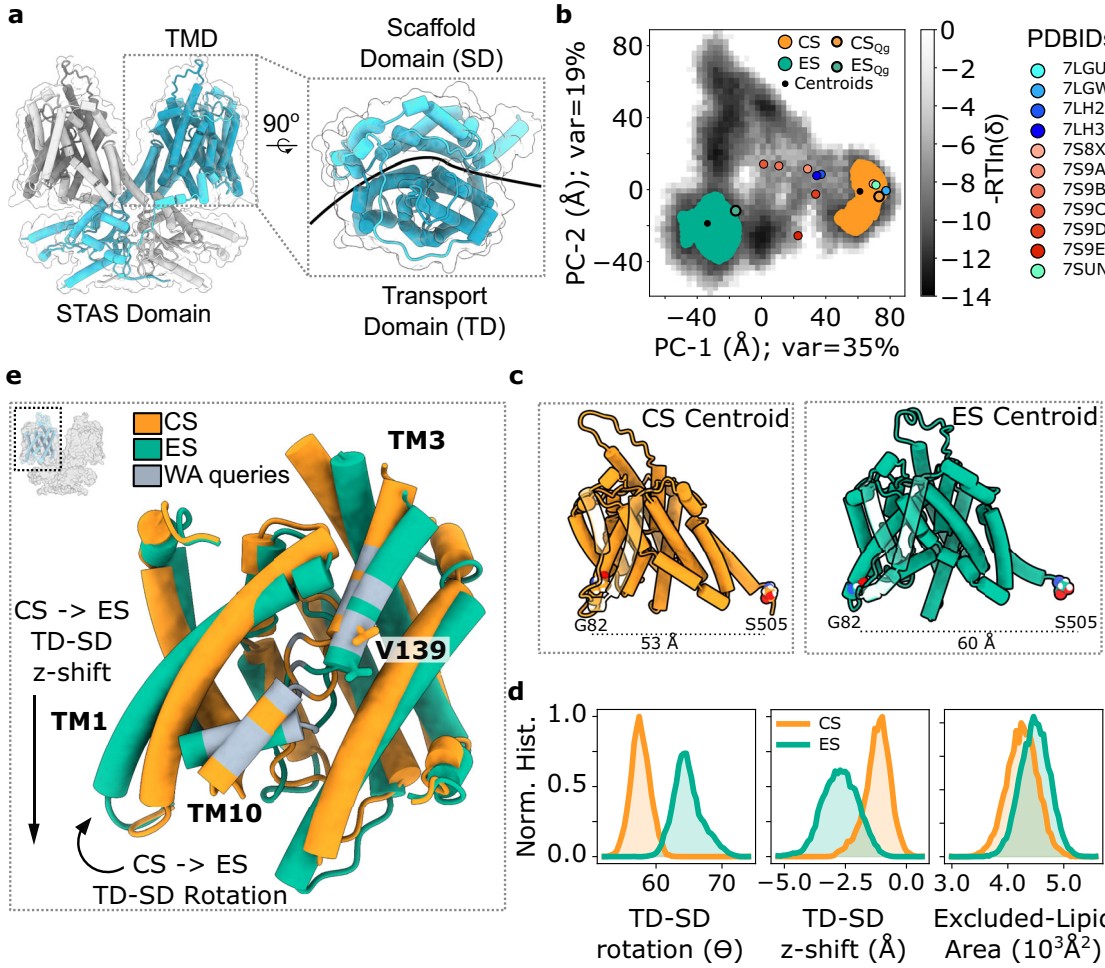

**Fig. 1 | Conformational landscape of prestin. a** Structural overview of the prestin homodimer, individual chains colored in grays or cyans to emphasize the domain-swapped dimer interface (left). Zoomed and rotated view of the transmembrane domain (TMD) from the extracellular side, dark and light cyan/gray indicate the Transport (TD) and Scaffold (SD) domains, respectively (PDBID: 7LGU).
**b** Conformational landscape of prestin visualized by principal component analysis (PCA) of equilibrium MD simulations. Frequencies are shown as -RTln(δ), where δ is the normalized count in the two-dimensional PCA landscape. Experimental cryo-EM structures of prestin in various conformations projected onto the PCA landscape in colored circles (see legend). Compact (CS) and expanded (ES) state clusters identified via DBSCAN are shown in orange and green, respectively. Centroids were calculated for each cluster and are shown in black. **c** Centroids of the CS and ES clusters represented in cartoon, showing the TMD of a protomer. G82–S505 distance for each centroid is displayed, demonstrating expansion of the TMD. **d** Distributions of structural observables: transport-scaffold (TD-SD) rotation (left), TD-SD z-shift distance (center), and intracellular protein-occupied area in the membrane, i.e., excluded-lipid area (right). Detailed approach for the lipid-excluded area found in the methods. **e** Overlay of the transport domains (aligned by scaffold domain) of the two centroids. ES displays a downward shift of (1.5 Å), and clockwise turn of -7°. Valine 139 shown as stick, other residues queried by SCAM/water-accessibility shown in gray. Structural context of analyses (inset). Source data are provided as a Source data file.

purified transporters are imaged in the absence of membrane potential. Based on conformations resolved by cryo-EM, Bavi et al. estimated charge movement of about 0.4 elementary charges between Up and Down-I states[31], substantially less than the ≈0.6–0.8 e⁻ derived from electrophysiological measures (e.g., refs. 33–36), indicating that these states may not represent the full extent of electromechanical reorientation. Moreover, all states except the 'Up' state required the presence of inhibitory anions, raising the concern that these structures represent inhibited conformations rather than intermediates of the electromotile cycle[32,37]. It was further pointed out that the reorientation suggested by the cryo-EM studies does not easily superimpose with the position of charged residues previously implicated in voltage sensing[32,33,37]. Thus, even if the observed structures represent native conformations, their relation to eM remains to be addressed.

To explore prestin's conformational landscape and dynamics under physiological ionic conditions and with control over the membrane potential we applied a combination of all-atom equilibrium molecular dynamics (MD) simulations, constant electric-field simulations, gating-charge calculations, and experimental biophysical approaches including cysteine modifications and voltage-clamp fluorometry (VCF).

MD simulations of prestin revealed a conformational landscape that goes beyond but includes previous experimental structures, and identified putative expanded and compact states. These conformations resembled inward-facing and occluded states of an anion transport cycle and transition predominantly involved a translational-rotational movement of the transport domain along the scaffold domain. Probing with cysteine-reactive compounds confirmed dynamic transitions between these states in electromotile mammalian prestin under physiological ionic conditions and identified an additional outward-facing state in transporting non-mammalian prestin. Finally, VCF with fluorophores attached to the transport domain allowed to directly monitor its translational dynamics. Importantly, VCF established that transport-domain movement is voltage dependent, precisely matching the voltage dependence of electromotility, thus tightly linking the elevator-like reorientation with eM.

These findings not only provide direct functional evidence for an elevator mechanism for SLC26 transporters, but also indicate that electromotility results from an elevator-like transport domain reorientation in prestin.

## Results

### Conformational landscape of prestin equilibrium simulations

To address the dynamic behavior of prestin, we performed atomistic equilibrium MD simulations over 6.2 μs, starting from the compact state (PDBID: 7LGU[19]), structurally consistent with the compact states found in independent cryo-EM studies[31,32], either with or without chloride ions in the binding pocket; all simulation systems were embedded in a POPC lipid bilayer and contained bulk [NaCl] of 200 mM. Prestin exhibited a stable homodimer for the duration of each respective simulation (Fig. 1a; Supplementary Fig. 1a, b).

Prestin explores a vast conformational landscape during equilibrium simulations as visualized by the first two principal components of PCA analysis (Fig. 1b; Supplementary Fig. 3d–i)—PCA reduces the complexity of the TMD conformation (high-dimensional: >2500 dimensions) to a set of orthogonal vectors which describe all correlated motions within the dataset. Principal components PC-1 and PC-2 account for 35% and 19%, respectively, of the total variance in TMD conformations. Two predominant and distinct conformational clusters of prestin were extracted from the simulations using the density-based spatial clustering of applications with noise (DBSCAN) algorithm of the conformational landscape defined by the first two principal components: (1) Compact state (CS), (2) Expanded state (ES). Conformations belonging to CS closely resemble the recently elucidated compact structure[19,31,32] (Fig. 1c).

In the ES cluster the transport domain is shifted and rotated relative to the scaffold domain (Fig. 1c–e). Notably, this clockwise turn (Fig. 1d, e) involves TM1 sliding outward from the transport-scaffold interface, increasing the width of the protein as monitored by the distance between Gly-82 of TM1 and Ser-505 of TM14 (Fig. 1c; Supplementary Figs. 1c and 2d) or the increased distance between centers of geometry of the transport and scaffold domains (Supplementary Fig. 1d). The expansion coincides with the binding pocket transitioning from an occluded state to an inward-open state, due to TM10 bending away from TM3 and exposing the anion-binding pocket to the intracellular solution (Supplementary Fig. 1e; Supplementary Fig. 2e, h). To quantify the expansion of the membrane upon transition from CS to ES, the area of excluded lipids was calculated for each frame, resulting in a 5.4% increase (~3 lipids) in the intracellular leaflet (Fig. 1d (right))—in contrast, the extracellular leaflet does not exhibit the same change in lipid-excluded area (Supplementary Fig. 1f).

We evaluated the conformational dependence of charge movement within the TMD. Charge movement was approximated by calculating the Z-component of the center of charge (Z-CC) (see "Methods"). The center of charge remains largely between 2 to 4 Å below the membrane center of mass (COM) (Supplementary Fig. 3c), with extremes in the CS and ES of 0 and 7 Å below the membrane COM, respectively. Thus, these data may indicate an outward movement of charge across the membrane upon ES–CS transition as expected from experiments (see below).

The landscape explored by the TM-domain of prestin encompasses that of the experimental structures of prestin[19,31,32], which cover multiple intermediate states between CS and ES (Fig. 1b; Supplementary Fig. 2). These data indicate that the molecular mechanism underlying prestin's electromotile cycle is not fully characterized by the current cryo-EM structures. Other smaller clusters identified appear to be structural intermediate states in-between the expanded and contracted conformations of prestin, and account for <11% of the total simulated data. CS and ES represent the extremes along the eM pathway and were thus chosen for all subsequent analyses.

In summary, the above MD analysis of prestin demonstrates the occupation and dynamic sampling of expanded versus compact states in the absence of inhibitory anions. The transition from CS to ES predominantly consists of a transport domain shift and rotation relative to the scaffold domain, coinciding with a reorientation of electric charges (Fig. 1d, e; Supplementary Fig. 3c), resembling the translational 'elevator' movement proposed to underlie anion transport by SLC26 transporters[27]. Notably, the findings support the idea that an elevator-like reorientation drives eM[38], prompting us to explore the reorientation experimentally.

### Accessibility mapping reveals an outward–inward transition common to electromotile and transporting prestin

We used substituted cysteine accessibility scanning method (SCAM) to experimentally explore the outward–inward dynamics of the transport domain seen in the MD simulations. We focused on TM10 and TM3, as inspection of the transport domain translation from CS to ES states suggested that residues along TM3 and TM10—the helices forming the anion-binding pocket—are exposed to the intracellular and extracellular environment in a conformation-dependent manner. Thus, the inward-open (ES) state, obviously predicts intracellular accessibility of positions within the binding pocket and along the helical TM10 lining the access funnel. Vice versa, we considered that positions of TM3, which is the topological counterpart related to TM10 according to the inverted symmetry of SLC26 proteins, may be preferentially accessible to the extracellular space in CS.

Experimentally, individual amino acids of TM3 and TM10 of prestin from rat (rPres[34]) were substituted with cysteine by site-directed mutagenesis. To avoid interference with native cysteines, all endogenous cysteines were replaced with alanine (rPres$_{\Delta Cys}$[20]). Most

cysteine substitutions introduced into rPres$_{\Delta Cys}$ were compatible with normal protein processing, membrane trafficking, and function, measured as eM-associated non-linear capacitance (NLC) by whole-cell patch-clamping (Supplementary Fig. 5a, b). NLC arises from the translocation of electrical charge that drives the electromotile reorientation and has been used widely as a robust proxy readout for electromechanical activity of prestin (e.g., refs. 2,39). Thus, irreversible inhibition of NLC by cysteine-reactive, membrane-impermeable MTS compounds, MTSES or MTSET, applied either from the extracellular or intracellular side (Supplementary Fig. 5c) indicated covalent modification and hence solute accessibility of the respective site[20,40].

All TM10 positions probed, including central binding site positions, are robustly modified by intracellularly applied MTSES and MTSET, but not by reagents applied from the extracellular side, indicating exclusive intracellular solute accessibility. These data have been reported previously[20] and are summarized in Fig. 2a (lower panel). To enable direct quantitative comparison of these experimental results with our MD simulations, water accessibility analysis was used as a quantitative proxy for solute accessibility. Per-residue water accessibility (WA) during MD simulations was monitored by counting the number of waters around a residue while tracking whether the water came from the intracellular or extracellular bulk solvents (see "Methods"). By this means, we determined WA of each residue from both extracellular and intracellular origin for the CS and ES clusters, respectively. The obtained in silico WA of TM10 residues revealed contact of these positions with water coming exclusively from the intracellular side (Fig. 2b, lower panel), in agreement with the experimental pattern of accessibility. With respect to conformational bias of accessibility, MD simulations indicated unhampered intracellular solute access of binding site positions (S398, L397, S396) in the inward-facing (extended) states but strongly restricted water access in the compact (occluded) state CS (Fig. 2b, lower panel; Supplementary Fig. 6). Hence the swift reactivity of the much bulkier MTS reagents at these positions provides evidence that extended states are occupied under native (physiological) conditions with Cl$^-$ as the principal substrate anion. Consequently, sampling of inward-facing states does not depend on the presence of inhibitory ligand anions that were required for the identification of equivalent extended conformations by cryo-EM[19,31].

Similarly, we probed TM3 for accessibility and found that TM3 amino acid positions facing the scaffold domain (G145C, L142C, V139C) were modified by extracellularly applied MTS reagents, whereas non-accessible positions point away from the scaffold domain surface (Fig. 2a, d–g; Supplementary Fig. 5d, e), in agreement with WA in MD simulations (Fig. 2b, upper panel; Supplementary Fig. 6). Notably, only a single position, namely V139, located close to the central binding site, was modified from both sides (Fig. 2a, g). Given the absence of a continuous permeation pathway (Supplementary Fig. 6p, q) that could allow access of the bulky MTSES in a single conformation, this finding immediately indicates alternating exposure of this position to the extra- and intracellular access pathways, thus attesting to two distinct conformational states (Fig. 2c). Again, these experimental findings match in silico accessibility. In the MD simulations V139 is the only position that clearly shows differential water access in ES and CS states, with access from the intracellular side in the extended state and extracellular access in the compact state (Fig. 2b, upper panel; Supplementary Fig. 6f). Thus, MTS reactivity strongly supports the dynamic reorientation between ES and CS states suggested by MD, and the similar Down and Up states, respectively, seen in the cryo-EM structures[19,31]. Notably, central binding site positions (TM10: L398C, L397C, S396C) that were readily modified from the intracellular side completely lacked access to extracellular MTS reagents (Fig. 2a[20]). Cysteine modification thus agrees with the pattern of access observed in the MD simulations across the entire conformational landscape, where negligible access of extracellular water into the binding site is observed (Fig. 2b). Hence, comparison of experiment with the MD simulation bolsters the completeness of the conformational landscape with respect to the transport-scaffold shifts observed in silico.

Together, these data provide a dynamic and quantitative view of the conformation-dependent solute accessibility in a process consistent with an elevator-like movement between ES and CS, however, conspicuously omitting a fully outward-facing conformation anticipated for active transporters (see below). This conclusion is consistent with the inability of extracellular anions to functionally interact with the binding site[18], and the absence of detectable anion transport rates[2,3].

Intriguingly, such an elevator process has been postulated as the anion transport mechanism of SLC26 transporters[25,27], although direct structural evidence is yet lacking. To address structural similarities between the putative eM dynamics and anion transport, we additionally probed transitions in an active SLC26 transporter. To this end, we focused on the SLC26A5 ortholog from zebrafish (zPres) which is closely related to electromotile prestin with respect to sequence conservation, but unlike the latter is an electrogenic anion transporter[2,3,41].

Again, helical segments of TM10 and TM3 were examined by SCAM. Following the previous approach, cysteine substitutions were made based on a zPres variant with most endogenous cysteines replaced with alanine (zPres$_{\Delta Cys}$; Supplementary Fig. 7a), which retained transport function (Supplementary Fig. 7b, c) and was insensitive to MTS reagents (Fig. 3a, c). Most cysteine substitutions introduced into zPres$_{\Delta Cys}$ were compatible with normal protein processing and transport function (Supplementary Fig. 7d, e), monitored as the electrogenic oxalate-chloride counter-transport current by whole-cell patch-clamping (Supplementary Fig. 7b)[3]. MTSES or MTSET were applied from either side and inhibition of transport current was used as the read-out for solute access to the respective cysteine-substituted position.

As shown in Fig. 3, the overall pattern of MTS modification in zPres was the same as obtained for mammalian prestin: residues along helical TM10 (positions 403–405) and in the binding pocket (S401C and M400C) were readily accessible to intracellular application of MTSES or MTSET. Positions in TM 10 beyond the binding site (A398C, T397C, V396C) were inaccessible, in accordance with location in the bulk of the transport domain. Thus, intracellular accessibilities reveal an inward-facing state, equivalent to the extended states observed in mammalian prestin. In TM3, positions oriented towards the transport domain-scaffold interface (V140C, L143C, and G147C) were accessible to extracellularly applied MTS reagents (Fig. 3a–e; Supplementary Fig. 8).

Strikingly different from mammalian prestin, however, binding site positions M400C and S401C (corresponding to positions L397 and S398, respectively, in rPres), were not only intracellularly accessible, but were also readily modified by extracellular MTSES (Fig. 3a, f and Supplementary Fig. 8). Notably, such extracellular accessibility is strictly incompatible with the experimental inward-facing structures of SLC26 homologs[21–25] and with experimentally determined conformations for mammalian prestin[19,31,32], all featuring an extracellular gate formed by tight apposition of transport and scaffold domains at the level of TM3. Consistently, MD simulations exhibited negligible water accessibility to these positions from the extracellular site of mammalian prestin (Fig. 2b; Supplementary Fig. 6) across the entire conformational landscape. Therefore, extracellular modification of the binding site positions directly indicates that unlike electromotile mammalian prestin, zPres alternates into an outward-facing conformation allowing extracellular solutes to interact with the substrate binding site (Fig. 3b). This conclusion is further supported by fast and complete modification of V140C (corresponding to position V139 in rPres) by extracellular MTSES and MTSET (Fig. 3e). Thus, susceptibility to MTS compounds is higher than found in rPres (cf. slower modification rates, Fig. 2g), consistent with exposure of the central

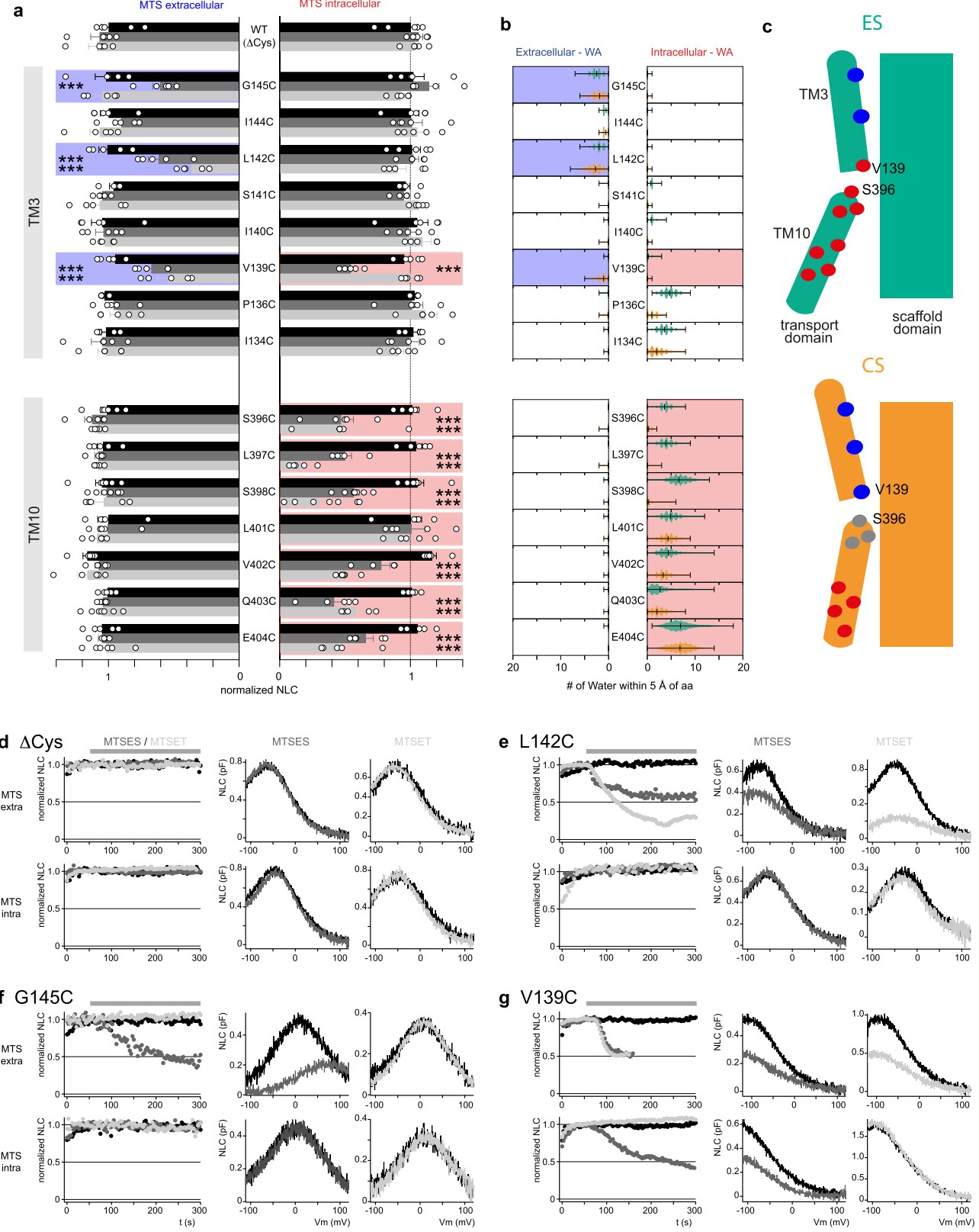

terminus of helix TM3 even to the bulky MTS compounds, as expected for an outward-open conformation that fully opens the binding site (Fig. 3b, lower panel).

Finally, we note that TM3 positions (S142C, L143C, G147C) were inaccessible from the intracellular side and TM10 positions were inaccessible from the extracellular side, albeit these positions were otherwise modified by MTS reagents (Fig. 3). Taken together, this

pattern provides (i) experimental evidence for transport dynamics with turnover between inward and outward-facing conformations alternately exposing inner and outer access pathways between transport and scaffold domain, lined by transport domain helices TM10 and TM3, respectively, and (ii) concomitantly the closure of an outer gate in the inward-facing conformation and an inner gate in the outward-facing conformation.

**Fig. 2 | Probing the conformational landscape by accessibility scanning.**
**a** Changes of peak NLC in response to application of MTS reagents (1 mM; MTSES, dark gray; MTSET, light gray; control, black) to the extracellular (left) or intracellular (right) side of cysteine substitution mutants expressed in CHO cells. Bars show residual NLC post-MTS reagent, normalized to signal before MTS application (±SEM; data from $n \geq 5$ independent cells, see Source data file). Significance level ***$p \leq 0.005$ (two-sided Dunnett's test). Upper panel, mutants in TM3; Lower panel, mutants in TM10. Coloration highlights positions sensitive to MTS reagents at the extracellular (blue) or intracellular (red) side. In mutant L401C, MTS reagents did not change NLC amplitude, but intracellular MTSET shifted voltage-dependence (Supplementary Figure 5f). **b** Extra- and intracellular water accessibility (WA) determined from MD simulations. Bars indicate average number of water molecules visiting each residue (100 ps/frame $n = 6$ MD replicas; error bars show SD within each respective cluster) originating from the extracellular solution (left) or intracellular solution (right). Orange, water contacts in the CS conformational cluster; green, contacts in the ES cluster. Positions exhibiting substantial water contact frequency consistent with experimental cysteine accessibility are highlighted in blue (extracellular WA) and red (intracellular WA). **c** Structural interpretation of experimental data. The extended conformation (ES, green) but not the compact state (CS, orange) is compatible with intracellular access to TM10 positions through V139 (TM3), including the anion-binding site centered approximately around S396. Corresponding experimental MTS reactivity thus confirms occupancy of ES under physiological conditions. Extracellular access of V139 at the lower end of helical TM3 is predicted only for the compact state (CS, orange). Thus, sensitivity of V139C to extracellular MTS demonstrates occupancy of CS. **d** Representative NLC recordings rPres with all endogenous cysteines replaced by alanines show insensitivity to MTS reagents applied either from the extracellular (upper panels) or intracellular side (lower panels). Left panels: Time course of NLC normalized to amplitude before application of MTSES (dark gray), MTSET (light gray), or control solution (black). Middle panels: representative NLC traces before (black) and after application of MTSES (dark gray). Right panels: representative NLC traces before (black) and after application of MTSET (light gray). **e–g** Representative recordings obtained as in (**d**) show inhibition of NLC selectively by extracellular MTS for L142C and G145C, but two-sided accessibility for V139. Source data are provided as a Source data file.

In summary, dynamics of the electromotile mammalian prestin and transporting zPres share the same transport domain-scaffold-reorientation—however, with the notable difference that mammalian prestin does not experience the outward-open state indispensable for full anion translocation.

## Sliding motion between transport and scaffold domains probed by cysteine cross-linking

We next aimed at experimentally testing the predicted translational (elevator) nature of the movement between transport and the scaffold domains. To this end, we employed a cysteine cross-bridging strategy, where linking of transport and scaffold domains would impair relative motion between both, but not other types of intramolecular dynamics. Amino acid pairs in TM8 (transport domain) and TM14 (scaffold), TMs that face each other as part of the transport domain-scaffold interface, were mutated to cysteines and $Cd^{2+}$ was applied as a cationic linker[42]. As expected, cysteine-free wt and single cysteine mutants in TM8 did not react to $Cd^{2+}$ (Fig. 4a, f; Supplementary Fig. 9a–d). However, unexpectedly the single Cys mutant D485C in TM14 displayed a strong reversible decrease of the NLC when extracellular $Cd^{2+}$ was applied (Fig. 4b, f, g; Supplementary Fig. 9e), suggesting cross-linking with a non-cysteine residue. In fact, the $Cd^{2+}$ effect on D485C was largely attenuated by removal of a proximate methionine residue, M143L in the opposing TM3 (Fig. 4c, f, g), indicating Cys-$Cd^{2+}$-Met cross-bridge formation between scaffold domain (D485C) and transport domain (native M143), and providing initial evidence for the relative translational movement between both domains.

Subsequent experiments further probing this movement with engineered cysteine pairs all included the M143L mutation. Double mutant D485C/V341C showed robust $Cd^{2+}$-dependent reduction of the NLC, (Fig. 4d, f), indicating $Cd^{2+}$-mediated cross-bridging that hampered the electromotile rearrangement. Similarly, cysteine pair D485C/V349C resulted in $Cd^{2+}$-sensitivity of NLC (Fig. 4e–g; Supplementary Fig. 9e). Further positions in TM8 (I344C, A345C, I348C, S352C, and V353C; Fig. 4g) were explored systematically in combination with D485C/M143C. These cysteine pairs were insensitive to $Cd^{2+}$ beyond the residual effect seen with D485C/M143C (Supplementary Fig. 9f). Although the α-carbons of the latter cysteine pairs are predicted with a distance sufficiently small to form a Cys-$Cd^{2+}$-Cys bridge (below 12 Å; refs. 43–45), the orientation of the side chains relative to the TM8/TM14 interface may explain the lack of $Cd^{2+}$ cross-bridging.

Together, $Cd^{2+}$ inhibition of NLC in two cysteine pairs between TMs 8 and 14 and one Cys-Met pair in TMs 3 and 14 supports a sliding, i.e., elevator, movement between transport and scaffold domains as an important component of the dynamics underlying NLC and hence eM.

In D485C/V349C, $Cd^{2+}$ also induced a leftward shift of $V_{1/2}$ (Fig. 4e, h), which is equivalent to a bias towards the ompact state[19,31]. Indeed, our MD simulations predicted the Cys-Cys distance of this pair to be smaller and hence preferable for $Cd^{2+}$-cross-bridging in CS (Fig. 4j), consistent with trapping this state by cross-bridging. Vice versa, the predicted D485-V341 distance is smaller for ES (Fig. 4i). In this cysteine pair, $Cd^{2+}$ cross-linking resulted in a smaller but significant positive shift, equivalent to a bias towards expanded states (Fig. 4h). This agreement between equilibrium MD simulations and effects of experimental cross-bridging further strengthened the idea that NLC and hence eM result from transition between ES and CS, and that this transition is voltage-dependent. Specifically, it indicates that CS corresponds to the compact ('short') state of prestin as suggested by its smaller lipid displacement, i.e., cross section (Fig. 1d (right)).

## Voltage dependence of transport domain dynamics

Finally, we directly addressed the involvement of the elevator dynamics revealed by equilibrium MD simulations, accessibility scanning, and cross-bridging in the generation of eM. The cardinal property of prestin-mediated mechanical activity is its voltage dependence. Here, we used voltage-clamp fluorometry (VCF) and constant electric-field simulations to directly probe for voltage dependence of molecular rearrangements[46–48].

We examined the transporter zPres first, because it can be readily expressed in *Xenopus* oocytes routinely used for VCF, as shown by large transport currents in the presence of sulfate. Of note, zPres also exhibits voltage-dependence, observable as voltage-dependent sensing currents or NLC[41]. Because equivalence of this electrical phenomenon in zPres to eM-related NLC in mammalian prestin has not been established, we first probed for hallmarks of electromotility, i.e., inhibition by salicylate and dependence on intracellular chloride[18,49,50]. As shown in Fig. 5a, NLC recorded from cells expressing zPres was readily inhibited by salicylate. Likewise, replacement of intracellular chloride abolished NLC (Fig. 5b). The recapitulation of eM-like sensitivity to salicylate and chloride suggested that zPres-mediated NLC reflects a structural transition similar to the dynamics underlying eM and thus provides a useful model system.

Given that transport domain translation was the dominant component of the dynamics observed in MD simulations, we attached an environmentally sensitive fluorophore, MTS-TAMRA, to the transport domain. To this end, we focused on the extracellular end of TM3, located at the membrane-solution interface. Mutant zPres variants with individual residues replaced by cysteine were expressed in *Xenopus* oocytes, TAMRA was covalently reacted to the cysteine, and fluorescence was recorded from voltage clamped oocytes (see "Methods"). Among several positions tested, zPres R151C uniquely

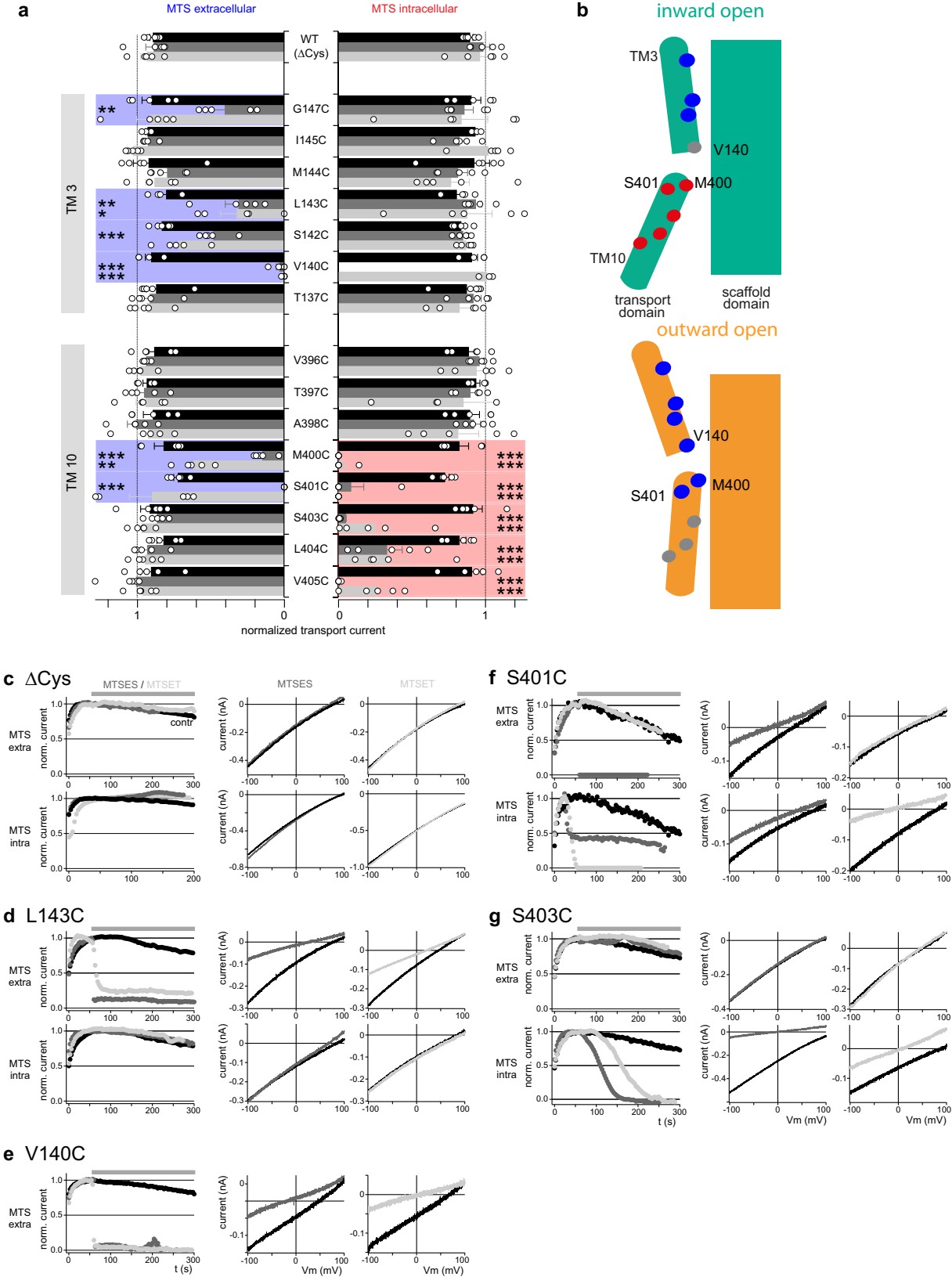

produced a TAMRA fluorescence signal that changed in response to imposed voltage steps (Fig. 5d, e; Supplementary Fig. 10a, b), thus reporting movement of the transport domain. NLC (Fig. 5c) and transport activity (Supplementary Fig. 10c–e) of zPres R151C was retained after TAMRA labeling. Voltage dependence of R151C-TAMRA fluorescence was sigmoidal and readily fitted with a first-order

Boltzmann function (Fig. 5e, f), consistent with the redistribution between two major states. Notably, voltage dependence of TAMRA fluorescence with a half-maximal fluorescence change at +97 mV matched the charge movement measured from TAMRA-labeled zPres-R151C in response to voltage steps (Fig. 5f; Supplementary Fig. 10f). This correspondence indicated that in zPres, voltage drives the

**Fig. 3 | Mapping inward-outward reorientations in the active transporter, zPres, by accessibility scanning. a** Relative changes of transport current in response to application of MTS reagents (1 mM; MTSES, dark gray; MTSET, light gray; control, black) to the extracellular (left) or intracellular (right) side of cysteine substitution mutants expressed in CHO cells. Upper panel, mutants in TM3; Lower panel, mutants in TM10. Shown are normalized currents from 5 independent cells per condition (±SEM). Significance level *$p \leq 0.05$; **$p \leq 0.01$; ***$p \leq 0.005$ (two-sided Dunnett's test). Coloration highlights positions sensitive to MTS reagents at the extracellular (blue) or intracellular (red) side. Source data are provided as a Source data file. **b** Structural interpretation of experimental solute accessibility. Two-sided accessibility of the two binding-side cysteine substitution mutants M400C and S401C indicates that zPres not only occupies states corresponding to ES (inward-open, green; upper panel) and CS that collectively allow for intracellular-only access to these sites. Experimental extracellular accessibility of these positions thus indicates occupancy of a fully outward-open conformation not observed in rPres (lower panel, orange). **c** Representative transport current recordings from CHO cells expressing zPres with most endogenous cysteines replaced by alanines (ΔCys; Supplementary Fig. 7a) show insensitivity to MTS reagents applied either from the extracellular side (upper panels) or intracellular side (lower panels). Left panels: Time course of current at 0 mV shown normalized to values after establishment of whole-cell configuration before application of MTSES (dark gray), MTSET (light gray), or control solution (black). Middle panels: representative whole-cell current traces in response to voltage ramps before (black) and after application of MTSES (dark gray). Right panels: representative current traces before (black) and after application of MTSET (light gray). **d, e** Representative transport recordings obtained as in (**c**) show inhibition of transport activity by extracellular MTS for TM3 residues L143C (**d**) and V14C (**e**). **f, g** Representative recordings obtained as in (**c**) show inhibition of transport activity by extra- and intracellular MTS for cysteine substitution S401C (**f**) and selectively by intracellular MTS application for S403C (**g**).

---

reorientation of the transport domain, which is physically associated with translocation of electrical charge across the membrane.

We further explored the movement detected by VCF in the presence of transport substrate, sulfate[3]. When sulfate was allowed to bind from the extracellular side, the voltage-dependence of the VCF signal shifted by $-39.5 \pm 4.1$ mV ($N = 17$) in a reversible manner (Fig. 5g–i). In alternating access transport (including elevator transport), the presence of substrate at only one side will bias the distribution between inward-facing and outward-facing states towards one side. If the same distribution also depends on voltage, then the effect of asymmetric substrate binding will be a shift in the voltage dependence of state distribution. The experimentally observed shift thus supports a model in which translation of the transport domain mediates transport and at the same time the reversible (capacitive) translocation of net electrical charge across the membrane.

The similarities between zPres and rPres with respect to NLC (Fig. 5a, b) support the idea that eM and associated charge movement in mammalian prestin likewise involve transport domain translocation. However, to address the dynamics underlying eM directly, we aimed at studying electromotile mammalian prestin. We therefore performed VCF with TAMRA attached to (mammalian) rPres at position R150C, homologous to R151C in zPres.

Voltage steps imposed upon *Xenopus* oocytes expressing this construct, labeled with MTS-TAMRA, resulted in fluorescence changes that were not observed with rPres lacking the cysteine at position 150 (Fig. 6a–d). VCF thus directly demonstrated voltage-dependent dynamics of prestin, and these dynamics involve the reorientation of the transport domain.

Strikingly, instead of recapitulating the voltage dependence of zPres-R151C-TAMRA, voltage dependency matched eM-associated NLC of rPres-R150C labeled with TAMRA, showing a $V_{1/2}$ of $-128.7 \pm 4.5$ mV and a slope (α) of $53.8 \pm 2.1$ mV ($n = 10$; Fig. 6e). For direct comparison, the differentiated VCF signal is plotted together with NLC ($V_{1/2} = -137.0 \pm 4.4$ mV; α $= 43.0 \pm 1.7$ mV; $n = 6$) in Fig. 6f. Moreover, the inhibitor of eM, salicylate, suppressed the voltage-dependent fluorescence signals of rPres-R150C-TAMRA (Fig. 6g, h). In accordance with previous measures of prestin's electromotile activity (i.e., NLC[18,51] or electromotility[51]), $V_{1/2}$ of the VCF dynamics was shifted to a more depolarized membrane voltage (Fig. 6i), consistent with salicylate changing voltage-dependent distribution between compact and extended states[51]. We observed that salicylate slightly quenched the fluorescence of TAMRA attached to rPres (Supplementary Fig. 11a). However, because fluorescence signals were normalized to fluorescence before each voltage step, quenching should not interfere with voltage-induced fluorescence changes. We confirmed this by measuring well-characterized VCF behavior of the voltage-sensitive phosphatase, Ci-VSP[52], where salicylate did not affect VCF amplitudes and did not shift voltage dependence as observed with prestin (Supplementary Fig. 11).

Taken together, the shared voltage dependence of R150C-TAMRA fluorescence and NLC and the inhibition by salicylate indicate that VCF directly reported the eM-generating molecular transition.

Considering the position of R150C-TAMRA, the VCF signals indicate that a translocation of the transport domain is involved in the electromechanical activity of prestin. Such reorientation appeared consistent with the CS-ES transition observed in equilibrium simulations as illustrated in Fig. 6c. Thus, comparison of ES and CS states suggests a substantial change of local environment experienced by position 150 along the conformational trajectory between these states (Fig. 6d). We therefore aimed to confirm that the charge movement observed with the ES–CS transition orthogonal to the membrane described above (Supplementary Fig. 3c) indeed reflects a voltage-dependent conformational change. To this end, we performed constant-electric field simulations in presence of membrane voltages (±200 mV) to estimate the gating charge associated with this conformational change. Simulation frames containing the CS and ES centroids (chain A) contain an independent chain B which hinders the decomposition of the gating charge to CS and ES specifically. Thus, we generated conformations representative of ES (ES$_{Qg}$, 1.65 Å) and CS (CS$_{Qg}$, 1.36 Å) clusters (shown in Supplementary Fig. 3a, c)–where the 'B' protomer for each respective simulation were restrained to identical positions–and subjected them to constant transmembrane voltages to analyze displacement charges, leading to a gating-charge ($Q_g$) estimate of $0.62 \pm 0.02$ e (Fig. 6j, k; see "Methods"). This calculated gating charge approaches experimental estimates for prestin's eM-associated charge translocation of about 0.6–0.8 e[33–36]. It is also consistent with the $Q_g$ estimate from Bavi et al.[31] (-0.4 e), which refers to the conformational transition between the cryo-EM structures only and therefore slightly underestimates the gating charge, as the conformations of the ES cluster are more expanded/downward shifted relative to the Down-I state (PDB: 7S9B) (Supplementary Fig. 2d)–thus highlighting the importance of additional conformational sampling by MD simulations. Importantly, these data predict a capacitive outward current upon ES–CS transition consistent with experiments.

## Discussion

Only recently, structural modeling[20] and experimental structures[19,31,32] allowed for the development of mechanistic models of eM at the protein level. Specifically, the recent series of single-particle cryo-EM studies reporting multiple distinct conformations of prestin collectively suggests an electromotile mechanism that involves transport domain-scaffold rearrangements and flexibility in scaffold domain TM6 leading to changes in the protein's cross-sectional area, and resultant state-dependent deformation of the lipid bilayer[37].

However, expanded states could only be observed in the presence of non-native ionic conditions, notably in the presence of inhibitory anions (salicylate, sulfate), whereas under physiological conditions, the electromotile conformational cycle occurs in the presence of

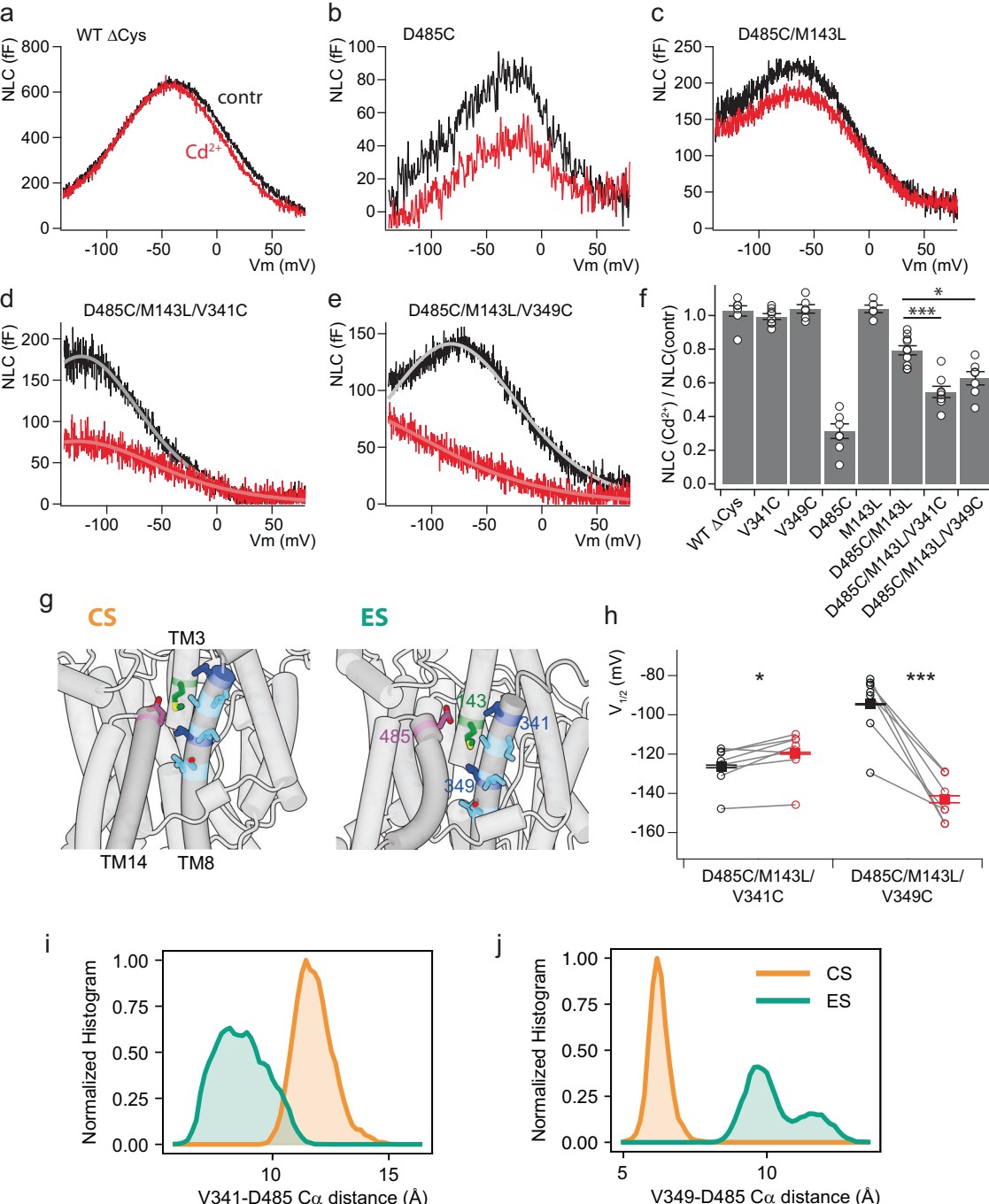

**Fig. 4 | Cysteine cross-bridging between transport and scaffold domains constrains eM-associated charge movement and ES-CS transition.**
**a–e** Representative NLC recordings before (black) and during application of Cd²⁺ (100 μM; red) onto CHO cells expressing the rPres mutants indicated. Continuous light curves in (**d**), (**e**) indicate fits of the derivative of a first-order Boltzmann function to the data. **f** Summary of Cd²⁺ inhibition, obtained from experiments as in (**a–e**). NLC in the presence of Cd²⁺ is shown normalized to NLC recorded before the application of Cd²⁺. Data from n = 7, 8, 6, 7, 5, 10, 8, and 7 independent cells expressing the respective mutants (± SEM). Significance level $p = 0.003$ (D485C/M143L/V341C vs. D485C/M143L), $p = 0.047$ (D485C/M143L/V349C vs. D485C/M143L), two-sided Dunnett's test. Source data are provided as a Source data file. **g** Structural view onto the transport domain scaffold domain interface highlighting the distances between residues in TM14, TM8, and TM3 examined for cysteine

cross-bridging in the CS (left) and ES (right) states. Cysteine-substituted positions sensitive to Cd²⁺ in combination with D485C (purple) are highlighted in dark blue, cysteine substitution without significant Cd²⁺ effects are shown in light blue. Position M143 is shown in green. **h** Changes in voltage dependence of NLC induced by Cd²⁺ in cysteine pair mutants (control, black; 100 μM Cd²⁺, red). Voltage at half-maximal charge transfer ($V_{1/2}$) was determined by fitting the derivative of a first-order Boltzmann function to NLC curves as shown in (**d, e**). Note that NLC delineates the distribution of prestin between compact and extended states, where equal distribution occurs at $V_{1/2}$ and depolarization favors contraction. Thus, a shift of $V_{1/2}$ towards negative potentials indicates an increase in occupancy of the compact state at any given potential. Data from the same n = 8 and 7 cells (±SEM) as in (**f**). **i, j** Distribution of distances (measured between $C_{alpha}$ atoms) between V349–D485 (**i**) or V341–D485 (**j**) for CS and ES clusters, respectively.

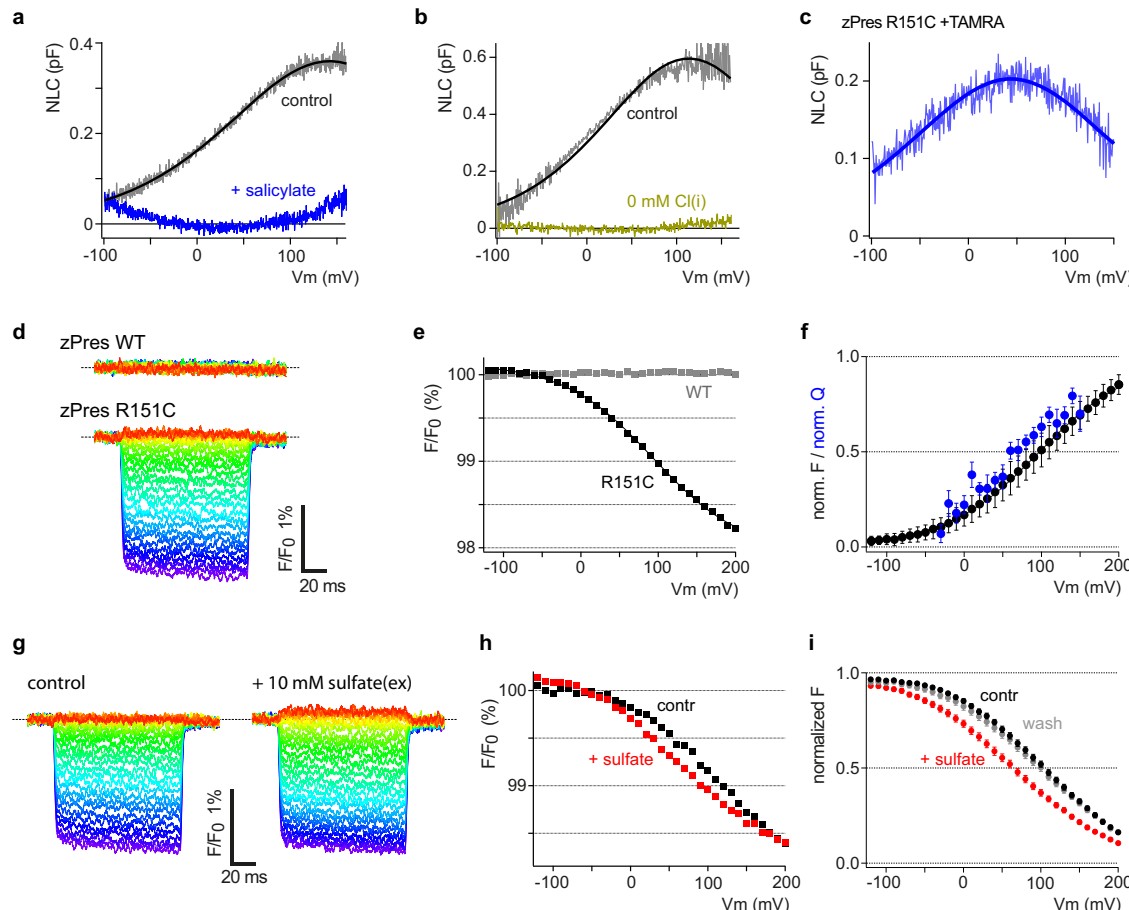

**Fig. 5 | Transport-associated transport domain movement in non-mammalian SLC26A5 detected by voltage-clamp fluorometry. a** Representative recordings of NLC mediated by zPres (recording from CHO cell transfected with zPres-GFP) before (black) and during application of salicylate (blue; 10 mM, extracellular). **b** NLC mediated by zPres requires intracellular Cl⁻. Subsequent NLC recordings were obtained from the same cell patched twice with different intracellular solutions containing either 160 mM Cl⁻ (black) or chloride replaced by an equimolar concentration of aspartate (olive). **c** NLC of zPres mutant R151C, labeled with MTS-TAMRA, recorded from a CHO cell. **d** Representative TAMRA fluorescence recordings from single Xenopus oocytes expressing either wild-type zPres or R151C mutant, each pretreated with MTS-TAMRA (see "Methods"). Oocytes were voltage clamped and subjected to voltage steps ranging from −120 mV (red) to +200 mV (purple) for 100 ms each (Supplementary Fig. 10a). Fluorescence is shown corrected for bleaching and normalized to mean before the voltage step. **e** Normalized fluorescence from recordings shown in (**d**) plotted as a function of membrane potential at each voltage step. Continuous line indicates a Boltzmann fit to the data ($V_{1/2} = 93.1$ mV; α = 57.5 mV). **f** Average fluorescence-voltage (F-V) relations from recordings from 17 independent oocytes (±SD) as in (**d**, **e**). Data were normalized to saturating fluorescence change as obtained from Boltzmann fits. For comparison, average Q-V curve recorded from zPres-R151C labeled with TAMRA is shown in blue ($n = 6$ independent CHO cells; ±SEM). Charge transfer was measured in response to voltage steps, isolated with a P/−8 step protocol (Supplementary Fig. 10f) and normalized to saturating charge obtained from Boltzmann fits to individual Q-V recordings. **g** Representative fluorescence recordings measured as in (**d**), either in the absence (left) or presence (right) of the transport substrate, sulfate (10 mM) in the extracellular medium. **h** F-V curves derived from the recordings shown in (**g**). **i** Average normalized F-V curves (±SEM) from $n = 17$ independent recordings as in (**h**) before application (black; $V_{1/2} = 103.7$ mV), during application of sulfate (red; $V_{1/2} = 64.3$ mV) and after wash-out of the substrate (gray).

the natural substrate, chloride[18]. Therefore, experimentally observed states may not reflect the physiologically relevant part of the conformational landscape[37]. Moreover, it has been argued that the position of charged residues that contribute to voltage sensitivity of prestin seem incompatible with the idea that prestin's voltage dependence results from translation of the transport domain normal to the membrane as suggested by the experimental structures[32,33]. Finally, all cryo-EM studies inevitably captured prestin at zero potential, and hence voltage dependence of the proposed transitions between observed conformations remains unknown. Together, these uncertainties warranted a functional approach towards a molecular model for electromotility.

To systematically address these open issues in the structural dynamics of prestin, we combined MD simulations and biophysical assays. MD simulations in near-physiological ionic conditions and in a POPC membrane environment indicate that prestin visits a complex conformational landscape that encompasses the structural snapshots previously identified in the experimental cryo-EM studies but reveals major conformations substantially beyond the cryo-EM structures. Thus, the electromotile cycle of prestin has not been covered comprehensively before. Two major conformational clusters differ in two main aspects: First, the cross-sectional area in the plasma membrane characterizes these states as compact (CS) and expanded (ES) states. Second, these major states differ in accessibility of the central anion-binding site. Whereas ES presents with the binding site readily accessible to the intracellular solution, defining an inward-open state, solute access of the binding site is blocked in CS, i.e., an occluded state in transporter terminology. The transition between both states is mostly mediated by a rotation of the transport domain relative to the scaffold domain along a trajectory normal to the membrane plane.

Given the distinct molecular dimensions of ES and CS with the former adopting a more expanded state, it appears plausible that

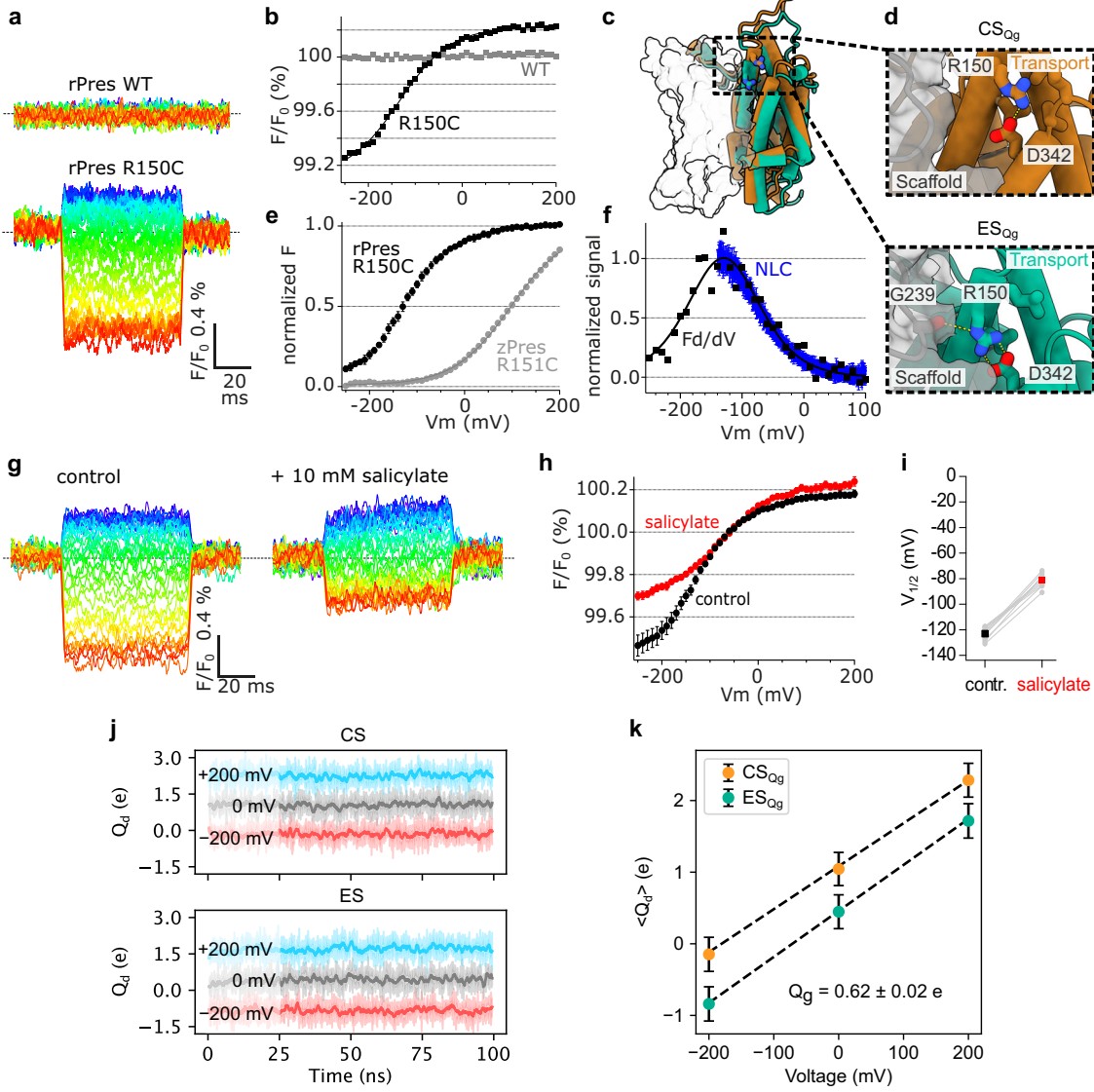

**Fig. 6 | eM-associated transport domain movement detected by voltage-clamp fluorometry. a** Representative fluorescence recordings from *Xenopus* oocytes expressing either wild-type rPres or R150C mutant, pretreated with MTS-TAMRA. Voltage-clamped oocytes were subjected to voltage steps ranging from −250 mV (red) to +200 mV (purple). **b** F-V curves derived from the recordings shown in (**a**). Continuous line indicates Boltzmann fit to the data ($V_{1/2} = −134.4$ mV, $\alpha = 54.5$ mV). **c** Predicted reorientation of R150 (side chain highlighted) between ES (green) and CS (orange). Side view; $ES_{Qg}$, and $CS_{Qg}$ are aligned by the relatively immobile scaffold domain (shown as surface rendering). **d** Zoom-views of R150 in the $CS_{Qg}$ (top) and $ES_{Qg}$ (bottom). Residues within 5 Å of R150's guanidinium group are shown as sticks. Hydrogen bonds/electrostatic interactions indicated by yellow dotted lines. **e** Average of F-V curves recorded from 10 oocytes (± SEM) as in (**d**, **e**). Data were normalized to saturating fluorescence change as obtained from Boltzmann fits to each individual measurement. For comparison, average F-V relation of zPres-R151C is replotted from Fig. 5f (gray). **f** Derivative of F-V curve from (**d**) shows

close agreement with eM-associated charge movement (NLC; blue). NLC was recorded from CHO cells (±SEM, $n = 6$ independent cells) expressing rPres-R150C prelabeled with MTS-TAMRA. **g** Representative fluorescence recordings measured as in (**a**), in the absence or presence of 10 mM extracellular salicylate. **h** Average F-V curves from 10 oocytes (±SEM) as in (**b**) before (black) and during application of salicylate (red). **i** Salicylate-induced shift of voltage dependence from measurements shown in (**g**). **j** Displacement charge ($Q_d$) for voltages −200, 0, and +200 mV. Traces visualized as windowed averages (1 ns window) over the mean of 3 replicates (dark) with SD (light). **k** Average charge displacement vs. voltage for CS and ES. Mean (± SD) of $Q_d$ calculated from pooled 25 ns simulation blocks ($n = 3$ independent replicate simulations each), with SD propagated from the individual replicates. $Q_g$ is calculated as the offset constant between the linear fits of prestin in the CS and ES state. Mean and standard deviation for $Q_g$ derived from linear fits of the 3 × 25 ns simulation blocks. Source data are provided as a Source data file.

ES-CS transitions mediate eM. Support for this view comes from the finding that the transition coincides with a gating charge translocation as revealed by constant electric field simulations, which directly links transmembrane voltage to the relative free energies of the CS and ES states[53] and thus predicts a direct impact of voltage on state distribution. The calculated charge movement of 0.62 e is reasonably close to the gating charge inferred from the voltage sensitivity of NLC or eM of typically around 0.6−0.8 e when measured from recombinant prestin[33–36] or 0.6−0.9 e in experimental data from native OHCs[49,54–56].

Two approaches were taken to validate the dynamics observed in our MD simulations. First, experimental solute accessibility along the predicted ion-access pathways located at the transport domain-scaffold interface and central binding site matched water contact frequencies obtained from simulations. In particular, the two-sided accessibility of the central position V139 strongly supports the inward-to-occluded transition that characterizes the major extended and compact states in MD. Second, to account for the possibility that other molecular reorientations might also result in similar patterns of

accessibility, we specifically aimed at testing the elevator-like transport domain/scaffold domain shift found in our MD simulations and recently implicated by discrete cryo-EM structures[19,31]. In fact, constraining relative motions between transport and scaffold domains by cross-linking at the domain interface is consistent with the elevator-like reorientation between CS and ES. With respect to electromotility, inhibition of the voltage-dependent NLC by cross-bridging indicates that those reorientations are likely voltage-driven and consequently that the transport domain-scaffold shift is involved in electromotility, which is generally thought to result from the same reorientation as NLC[11,13]. Of note, the observed shift of voltage-dependence by cross-linking experimentally supports the idea that CS depicts the compact ('short') state of prestin consistent with its smaller molecular diameter in the membrane plane (Fig. 1c (left)).

Finally, a direct demonstration of voltage dependence is provided by the VCF approach that combines a fluorescent sensor for conformational changes with control of membrane potential in the native environment. Since R150 is located at the outer membrane surface outside the membrane electrical field (Fig. 6c, d), its voltage-driven reorientation indicates movement of a larger structural unit, which involves transfer of electrical charge across the membrane electrical field, i.e., with a substantial vectorial component normal to the membrane. As R150 is located within TM3, VCF demonstrates a transport domain reorientation that precisely shares the voltage sensitivity of NLC and electromotility as well as sensitivity to the eM inhibitor, salicylate, thereby tightly linking a transport domain movement to prestin's electromechanical activity. One issue to consider is whether VCF reports on the same ES−CS reorientation identified by MD simulations and mapped by SCAM and cross-linking. Although VCF does not directly reveal the direction of the movement of the labeled position, the environmental sensitivity of TAMRA fluorescence and its position at the outer membrane interface suggest that a major component of the detected movement is orthogonal to the membrane, a key feature of the ES−CS transition.

Movements at the intracellular face of the protein have been suggested to contribute to eM. Thus, pronounced bending of the intracellular end of TM6 was observed in a partially salicylate-bound structure[31]. Also, charged residues that may contribute to voltage sensing were found predominantly close to the intracellular surface of transport and scaffold domains[33], which led to the suggestion that local rearrangements of these structures may mediate eM[32]. However, monitoring of the eM-associated physical movement by VCF at the extracellular side of the transport domain (position R150C) provides clear evidence that a bulk movement of the transport domain is associated with eM. The pivotal role of transport domain reorientation for eM is further strongly supported by the finding that constraining it by cross-linking hampers NLC. Nevertheless, additional conformational flexibility, including TM6 bending and flexibility within the transport domain[19] may contribute to electromechanics of prestin. In particular, distortion of the lipid bilayer elicited by the gross conformational changes along the ES−CS trajectory is probably important for electromechanical output at the membrane level[19,31,32].

Altogether, the most parsimonious conclusion from the combined MD simulations and molecular biophysical experiments is that eM arises from a conformational transition between ES (inward-open) and CS (occluded) states, with a major component of this movement being the elevator movement of the transport domain relative to the scaffold domain.

The origin of prestin's voltage sensitivity has been controversial, with initial suggestions proposing the bound substrate anion as an 'extrinsic' voltage sensor[18], while others argued in favor of a delocalized 'intrinsic' sensor provided by charged protein residues[33,57,58]. The present results show that the bound anion is located more towards the extracellular side in the contracted CS conformation. Since depolarization drives prestin into the compact state, the anion moves opposite

to its own electrical driving force and hence cannot provide the sensor charge. Consequently, the overall charge movement must be afforded by the sum of intrinsic electrostatic charges distributed across the apoprotein and the anion cradled in the central binding site. In other words, both must act together as a 'hybrid' voltage sensor, where, ironically, the anion acts to reduce the net charge in counteracting part of the intrinsic sensor charges. Since sensor charges must move across the electrical field, i.e., normal to the membrane plane, most sensor charges are likely provided by the transport domain that undergoes the elevator translocation. Indeed, our simulations demonstrate a conformation-dependent reorientation of the charged residues along the membrane normal, with a gating charge of 0.62e and a predicted outward current upon ES−CS transition, consistent with experiments (Fig. 6; Supplementary Fig. 3c).

With availability of the comprehensive conformational landscape revealed by our MD simulations, it will now be possible to study details of the dynamics of the distinct proteins' domains, to fully calculate the charge movement, and to identify all components of the spatially distributed voltage sensor. Specifically, the role of Cl⁻ binding in voltage sensing and charge movement is unclear at present and requires further study.

Comparison of various experimental structures of SLC26 and related (SLC4 and SLC23) transporters have led to the idea that anion transport in the SLC26 family is mediated by an elevator mechanism[27], although definitive evidence for this structural mechanism has been lacking. Only very recently, an outward-facing conformation has been reported for SLC26A4[59]. Structural and functional similarities between mammalian and transport-active non-mammalian prestin isoforms have previously led us to speculate that electromotility results from a modified, incomplete anion transport cycle[2,20,37]. Most obviously, this idea is supported by the fundamental observation that binding of monovalent anions[18,39,60]—most likely to the canonical binding site[19,20,31]—is the prerequisite for electromotile activity. Our present data now substantiate this idea with direct evidence: both functional experiments and MD simulations show an elevator-type movement as the molecular movement underlying eM, which includes a partial transmembrane translocation of the anion bound into the transport domain's binding site.

Given the lack of experimental structural data for the full transport cycle of SLC26, we also probed into the dynamics of an SLC26 transporter, drSLC26A5 (zPres), closely homologous to mammalian prestin. Thus, cysteine accessibility scanning (SCAM) indicated that transporting SLC26A5 undergoes a reorientation between transport and scaffold domains congruent to that of mammalian electromotile prestin; however, with the important difference of full opening towards the extracellular space, therefore mediating alternating access of the central binding site. Moreover, by using VCF, we identify a transport domain reorientation that shares all hallmarks of the eM-associated rearrangements mapped in mammalian prestin: (i) the position of the VCF reporter fluorophore indicates a movement component normal to the membrane. (ii) movement is voltage-dependent (although with a strongly right-shifted voltage dependence). (iii) the reorientation shares functional properties with eM in terms of anion dependence and salicylate sensitivity. This reorientation is indeed involved in anion transport as indicated by its sensitivity to the presence of the divalent transport substrate, sulfate.

Taken together, we conclude that the voltage-dependent step that in mammalian prestin generates eM, is structurally homologous to the elevator movement mediating transport in SLC26 homologs.

Accordingly, the fundamental principle underlying eM was not acquired de-novo during emergence of the mammalian line, but evolved by gradual modification of a transport mechanism[61,62]. Along with the emergence of electromotile function, prestin's transport activity was lost by disabling the transition into outward-facing states, possibly to avoid compromising the hair cell's ionic homeostasis by

associated ion fluxes. At present it is unclear which structural peculiarities of mammalian prestin prevent the transition to a full outward-open state. Vice versa, why are non-mammalian SLC26A5 orthologs not mechanically active, if the underlying molecular reorientation is essentially the same? Interestingly, data from chicken hair cells suggest that also non-mammalian SLC26A5 may drive eM[63]. Electromechanical activity (of non-mammalian SLC26A5) may just be smaller and difficult to detect, because it occurs at quite unphysiological positive membrane potentials with a very shallow voltage dependence. Moreover, the non-mammalian homologs may not reach the excessively high densities observed in mammalian OHCs that support robust and easy-to-detect cellular mechanical events. This view emphasizes the fundamental nano-mechanical nature of ion transport by SLC26 solute transporters, but evolutionary optimization of this principle shaped mammalian prestin via acquisition of a physiologically meaningful voltage dependence and by kinetic optimization to comply with acoustic requirements. Nevertheless, distinctive differences in the nature of the molecular dynamics, such as enhanced flexibility in TM6[31] or increased distortion of the lipid bilayer[19,31,32], may also be responsible for an outstanding electromechanical potency of mammalian prestin. More structural information on multiple states of distinct SLC26 transporters will be required to answer these questions.

## Methods

### MD simulations

**Construction of molecular simulation systems.** All-atom molecular dynamics (MD) simulations of human SLC26A5 (prestin) were built from the cryo-EM structures of the chloride bound, compact state (PDBID: 7LGU[19]). Missing atoms and terminal caps were added using the *psfgen* utility in visual molecular dynamics version 1.9.3[64]. The final prestin-model residue range is from 13–580, 614–725, excluding a large segment of the STAS intervening sequence. Experimentally resolved cholesterol and lipids were removed from the MD model. The N- and C-termini were neutralized by acetylation (ACE) and N-methylamidation (NME), respectively. Default protonation states for all residues were chosen, except for D653 which was previously determined to be protonated[19]. The proteins were embedded in a pre-equilibrated (i.e., solvated and ionized) 1-palmitoyl-2-oleoyl-glycero-3-phosphocholine (POPC) bilayer using 'gmx mdrun -membed' (using these parameters: nxy=1000, nz=0, xyinit=0.5, xyend=1.0, zinit=1.0, zend=1.0, rad=0.22, ndiff=0, max-warn=0, pieces=1, asymmetry=yes). Two simulation systems were generated, Cl⁻ bound (PCl) and Cl⁻ Free (PClF), where the experimentally resolved Cl⁻ are removed. The molecular composition of the simulation systems is provided in Supplementary Table 1. All-atom simulations were conducted using the GROMACS software package, version 2021[65], with protein atoms being defined by the CHARMM36m force-field. Ions and lipids were described using default CHARMM36 parameters, and the TIP3P model was used for waters—an established force-field used for membrane-protein simulations[66,67]. An integration time step of 2 fs was used for all simulation steps. All covalently bound hydrogens were constrained using LINCS[68]. Nonbonded Van der Waals interactions were calculated using the Lennard-Jones potential with a cutoff radius of 1.2 nm; forces were smoothly switched off in the range of 1.0–1.2 nm. Electrostatics were calculated using the smoothed particle-mesh Ewald[69] method with a real-space cutoff distance of 1.2 nm. The PCl system was minimized using a steepest decent protocol. Following the minimization, PCl was equilibrated in the isothermal-isobaric ensemble (NPT) using a Berendsen barostat[70] in three subsequent steps: (1) position restraints on all protein heavy atoms, and the z-component of lipids, (2) position restraints on all protein heavy atoms, (3) position restraints on protein backbone heavy atoms for an additional 50 ns. System equilibration was determined by monitoring the number of protein atoms which come into contact (i.e., distance < 5 Å) with atoms of the lipid headgroup for 140 ns (system converged within 50 ns, Supplementary Fig. 4) during the 2nd step of equilibration (all heavy atom restraints).

For PClF, chlorides were removed after the 2nd equilibration step (all protein heavy atoms constrained), and the 3rd equilibration step was performed for an additional 50 ns. Three snapshots—each separated by 10 ns—were taken from the 3rd equilibration step from PCl and PClF, respectively, containing positions and velocities, and were used as initial points for production simulations. Production simulations were carried out in the isobaric ensemble (NPT) using the Parrinello-Rahman barostat[71].

**Analysis of the conformational landscape of prestin.** To evaluate the conformational landscape explored by prestin in the equilibrium simulations, we employed principal component analysis (PCA). PCA is an unsupervised dimensionality reduction method which decomposes complex high-dimensional data into eigenvectors which describe correlated motions, the corresponding eigenvalues report on the total variance encompassed by each eigenvector (i.e., correlated motion). The conformational analysis was focused on the TM-domain of prestin. To minimize computational memory costs, we reduced the total number of residues within the TMD, excluding loops (i.e., including residues 76 to 505; 350 residues), by using every 5th residue of the 350, starting with 76 (i.e., 76, 81, 86, etc.) providing a total of 76 residues. The inter-Cα distances within the subset of TM residues were used as input features for PCA. The reported experimental structures[19,31,32] were mapped onto the first two principal components to visualize the extent of the conformational sampling in the equilibrium simulations alone (Fig. 1b)—the 1st and 2nd principal components (PC-1 and PC-2) contain 35% and 19% of the explained variance, respectively. DBSCAN clustering of the first two principal components – DBSCAN parameters: eps = 2 and min_samples = 150 – generated seven clusters across the conformational landscape. Five of the seven clusters encompassed 11% of all simulations, 47% of the simulations were relegated to an 'unassigned cluster' (data not shown). The two remaining clusters encompass 42% of all sampled conformations (Fig. 1b), localized to the extremes of PC-1, and were thus selected for further analyses.

Distributions of structural observables for the selected clusters reveal two distinct conformations of prestin, in the CS (17% of total data) and a putative ES (25% of total data) (Fig. 1). The shift and rotation of the transport domain relative to the scaffold domain are hallmarks of the elevator mechanism, and were thus quantified in the following manner. We calculate the vertical translation of the transport domain (residues 76–151, 173–196, 332–436) center of geometry (COG) relative to the scaffold domain (residues 207–305, 436–507) COG. Additionally, we calculated the angle between the transport and scaffold domain, using the COGs for the bottom halves of each domain (transport domain: bottom residues 76–93, 117–137, 185–196, 351–380; scaffold domain: bottom residues 207–216, 272–292, 449–507), with the upper half of the scaffold domain defined as the hinge (scaffold: upper residues 217–271, 293–305, 437–448) (Fig. 1d). Representative conformations of the two clusters were extracted for visualization. We calculated cluster centroids which minimize the pairwise distance with each conformation in the cluster according to their RMSD—this method finds the centroid directly from conformational similarity—resulting in two conformations whose Gly82 to Ser505 Cα-distance were 53 Å and 60 Å for CS and ES, respectively (Fig. 1c).

**Analysis of cross-sectional area via excluded-lipids.** Prestin's physiological function of cochlear amplification through expansion/contraction of the OHC is driven by a quasi-piezoelectric area motor. Previous attempts to computationally characterize the cross-sectional area have used low-resolution 3D masks of the protein and arbitrary isodensity-values to approximate the cross-sectional area—while these approximations work in a comparative manner, they lack a physical basis. Thus, it is imperative to have an objective means of quantifying the expansion/contraction of prestin in silico.

Here we attempt to ameliorate this issue by measuring, instead, the effects of the prestin on the lipid bilayer. At each frame, the area-per-lipid (APL) in the bulk-membrane (i.e., at >70 Å distance from the protein center) is calculated for the intra- and extracellular leaflets, i.e., total 'bulk' area divided by the number of 'bulk' lipids. From the bulk APL we calculate the theoretical number of lipids, which represents the number of lipids that would be in the membrane in the absence of an embedded protein. The difference between the theoretical and actual number of lipids reveals the area occupied by the protein, i.e., the 'excluded-lipid area' (Fig. 1d (right); Supplementary Fig. 1f). We demonstrate here a means of calculating the process of expansion-contraction directly from all-atom equilibrium MD from well-established lipid-calculations.

**Analysis of water accessibility via history labeling.** Per-residue water accessibility was used as a proxy for experimental scanning cysteine accessibility method (SCAM). For the residues queried, the number of water-oxygens within 5 Å of the residues backbone atoms (N-Cα-C) were counted every frame (100 ps). To determine whether the counted waters came from the intracellular (*intra*) or extracellular (*extra*) environment, a state-vector was recorded and tracked the last visited bulk-phase (*intra* vs. *extra*). Per-residue water accessibility was mapped onto the first two principal components, and clustered according to the DBSCAN states (CS, ES) (Fig. 1b; Supplementary Fig. 6a–o). State-dependent water accessibilities of the residues were calculated and visualized with violin plots showing the mean +/− a standard deviation (Fig. 2b).

**Chloride binding analysis.** Chloride is an obligate anion for electromotility (eM) in mammalian prestin, and thus understanding the conformation dependence on Chloride binding is imperative to understanding the eM mechanism. The chloride bound state was assessed by monitoring the distances between a chloride atom and the anion-binding pocket, defined by the closest residues to the bound chloride in 7LGU, F137, S396, and S398. Chlorides are unbound (0) if they are more than 7 Å from the Cα of S396 or S398, partially bound (1) when within 7 Å of the Cα of S396 and S398, and bound (2) when within 7 Å of the Cα of F137 and S396 and S398. The average chloride bound state is mapped to the conformational landscape (Supplementary Fig. 3b). Overall, prestin maintains chloride binding for values of PC-1 > 0, with exception to the first ~50 ns of the PClF simulations.

**Center of charge calculations.** We evaluated the distribution of charges within the membrane by calculating the z-component of the charge center (CZ) within the membrane (Eqs. (1)–(3)).

$$CZ = CC_{TMD} - MC_{MEM} \qquad (1)$$

$$CC_{TMD} = \frac{\sum_i |q_i| z_i}{\sum_i |q_i|} \qquad (2)$$

$$MC_{MEM} = \frac{\sum_i m_i z_i}{\sum_i m_i} \qquad (3)$$

Where $CC_{TMD}$ is the charge center of the TMD, $MC_{MEM}$ is the mass center of the membrane, $|q_i|$ is the absolute partial charge of an atom of any charged residue within the TMD (+6 e), $m_i$ is the mass of all phosphate atoms, $z_i$ is the z-coordinate of the atom in the unaligned simulation-box. We visualized the conformation-dependent charge distribution by coloring the conformational landscape with the mean CZ (Supplementary Fig. 3c).

**Constant electric-field simulations.** Non-equilibrium simulations were performed to estimate the gating charge ($Q_g$) corresponding to the conformational change between extreme states of prestin along the electromotile cycle (i.e., CS and ES). $Q_g$ calculations rely on system-level information (e.g., charge displacement), and thus complicates the decomposition of $Q_g$ to individual protomers of a multimeric protein. Prestin rapidly loses its symmetry during simulation, exhibiting independent sampling of the two protomers at the simulated timescales. To ensure that we calculate $Q_g$ between the ES and CS states specifically, we ran additional equilibrium simulations, where one protomer of prestin has all atoms restrained. These simulations were initialized from PClF (PDB: 7LGU without Cl⁻) and were run under the same conditions as the previous production simulations—4 replicates were run for ~500 ns each. Frames were selected, which minimized the RMSD of the free protomer to the CS (1.36 Å) and ES (1.65 Å) cluster centroids, respectively. Whereas the frame selected for the CS ($CS_{Qg}$) is located near the CS-centroid in PCA space, the frame selected for ES ($ES_{Qg}$) sits near the outer edge of the ES cluster, potentially under-estimating the gating charge between the CS and ES centroids (Supplementary Fig. 3a, c). These frames were then subject to further non-equilibrium constant electric-field (CE-Field) simulations, in the presence of heavy atom restraints for the entire protein (1 kJ/mol nm²) —thus the resulting $Q_g$ must be exclusively due to the transition between CS and ES of the selected protomer. CE-Field simulations were performed with field strengths of −0.013, 0, and +0.013 V/nm, corresponding to +/−200 mV, with 3 replicates for 100 ns each (i.e., 300 ns/voltage/system). As preparation for $Q_g$ calculations, simulations were re-centered and unwrapped according to Kostristkii and Machtens[72]. Though chlorides frequently interact with the ABP in equilibrium simulations, no chlorides interact with the anion-binding pocket of either protomer throughout CE-Field simulations due to steric occlusion of restrained side chains.

**Calculation of the gating charge.** Here we employ the Q-route method for calculating $Q_g$[53], which evaluates the difference in the average displacement charge for a given state and voltage $\langle Q_d \rangle_{s,V}$,

$$\langle Q_d \rangle_{s,V} = \left\langle \sum_{i=1}^{N} q_i \frac{z_i^u}{L_z} \right\rangle_{s,V}, \qquad (4)$$

where $\langle Q_d \rangle_{s,V}$ is evaluated as the time-averaged mean over the sum of the product of partial charges $q$ and unwrapped position along the membrane normal $z^u$ for each atom $i$. The displacement charge varies linearly according to $\langle Q_d \rangle_{s,0} + CV$, where $C$ is the system capacitance, which would be roughly identical between the two states. Finally, $Q_g$ is evaluated as the offset constant between the two linear relationships.

$$Q_g = \langle Q_d \rangle_{CS,0} - \langle Q_d \rangle_{ES,0} \qquad (5)$$

The last 75 ns for each simulation were used to create 3 × 25-ns blocks used for averaging. Each block was averaged and linearly fit to produce a mean and standard deviation for the calculation of $Q_g$ (Fig. 6k). Error bars for the $\langle Q_d \rangle$ vs. voltage plot come from error propagation of the block-averaged trajectories (Fig. 6k).

## Molecular biology

cDNAs coding for SLC26A5 from *Rattus norvegicus (rPres)* and *Danio rerio* (zPres) were cloned into pEGFP-N1 expression vector (Clontech Laboratories, Mountain View, CA, USA), yielding C-terminal GFP fusion constructs described previously[2,3]. For expression in *Xenopus* oocytes, rPres or zPres were cloned into the pGEMHE vector. Here, zPres constructs contained a functionally silent R505K mutation and rPres constructs contained a 6xHis tag at the C terminus. Point mutations were introduced by standard primer mismatch mutagenesis using QuickChange Site-Directed Mutagenesis Kit (Agilent Technologies, Santa Clara, CA, USA). All constructs were verified by sequencing.

To synthesize RNA for injection into oocytes, plasmids were linearized with NheI and transcribed using the T7 mMessage mMachine kit (Thermo Fisher Scientific, Waltham, MA, USA). All plasmids can be obtained from the corresponding authors upon request.

## Patch-clamp electrophysiology

**Patch-clamp recordings.** NLC was recorded from CHO cells (CRL-9096, American Type Culture Collection) transiently expressing the respective prestin constructs[2]. Plasmid DNA was transfected into Chinese Hamster Ovary (CHO) cells using JetPEI transfection reagent (Polyplus, Illkirch, France). For whole-cell patch clamp, cells with unequivocal membrane fluorescence were selected 24–48 h after transfection. Recordings were carried out at room temperature (20–22 °C) with EPC10 amplifiers (Heka, Lambrecht, Germany) controlled by Patchmaster software (Heka).

The experimental bath chamber was continuously perfused with a solution containing was (in mM): 144 NaCl, 5.8 KCl, 1.3 CaCl$_2$, 0.9 MgCl$_2$, 10 4-(2-hydroxyethyl)piperazine-1-ethanesulfonic acid (HEPES), 0.7 Na$_2$HPO$_4$, 5.6 glucose (pH adjusted to 7.4 with NaOH). Salicylate was added to the extracellular solution at 10 mM (Fig. 5a). For exchange of extracellular solutions, cells were locally perfused using a gravity-fed local application pipette.

Patch pipettes were filled with an intracellular solution containing (in mM): 160 CsCl, 1 HEPES, 1 K$_2$EGTA (pH 7.3 with CsOH). For measurements with nominally Cl$^-$-free conditions pipettes were tip-filled with solution containing 160 mM Cs-aspartate instead of 160 mM CsCl and then back-filled with Cl$^-$-containing solution to ensure stable electrode offset.

**Non-linear capacitance (NLC).** Whole-cell membrane capacitance ($C_M$) was recorded using the sine+DC software lock-in function of Patchmaster utilizing 2 kHz stimulus sinusoids. Voltage-dependent NLC was assessed by recording $C_M$ during voltage ramps as described and plotted as a function of membrane potential ($V_M$). NLC was quantified by fitting the derivative of a first-order Boltzmann function to the $C_M(V_M)$ traces,

$$C_M(V_M) = C_{lin} + \frac{Q_{max}}{\alpha e^{-\frac{V-V_{1/2}}{\alpha}}\left(1 + e^{-\frac{V-V_{1/2}}{\alpha}}\right)} \quad (6)$$

where $C_{lin}$ is linear membrane capacitance, $V_M$ is membrane potential, $Q_{max}$ is maximum voltage-sensor charge moved through the membrane electric field, $V_{1/2}$ is voltage at half-maximal charge transfer and $\alpha$ is the slope factor of the voltage dependence. The amplitude of NLC was quantified as peak NLC, $NLC_{max} = C_M(V_{1/2}) - C_{lin}$.

Charge transfer in response to voltage steps (Fig. 5f) was measured using a P/−8 protocol[41]. Charge was obtained by integration of the averaged transient currents (Supplementary Fig. 10f) over time and was plotted versus membrane potential. Charge-voltage plots were fitted with a two-state Boltzmann function to obtain voltage at half-maximal charge transfer, $V_{1/2}$.

**Transport currents.** For transport measurements, intracellular solution contained a high concentration of the divalent transport substrate, oxalate (in mM): 106 Cs-oxalate, 10 NaCl, 10 HEPES; pH 7.3 with CsOH. Transport currents were recorded in response to voltage ramps (−120 to +120 mV). For quantification, currents at 0 mV were extracted from current responses, thus minimizing impact of leak currents, and currents were normalized to stable currents reached after equilibration of cytosolic concentrations with pipette solution.

**Cysteine modification (SCAM).** Stock solutions (0.2 M) of (2-sulfonatoethyl)methane-thiosulfonate (MTSES) or [2-(trimethylammonium)ethyl]methane-thiosulfonate (MTSET) were prepared in DMSO

and stored frozen at −20 °C until use[20]. MTSES or MTSET were added to intracellular or extracellular solutions immediately before use. Final concentration of DMSO in intra- or extracellular solutions was 1% or less.

**Cysteine cross-linking experiments.** Prior to measurement, cells were incubated with 5 mM DTT solution for at least 15 min at 37 °C to reduce any spontaneously formed disulfide bonds. Extracellular solution containing 100 μM CdCl$_2$ was applied for 180 s and washed out for another 180 s. Solution exchange was performed using an VC3-8xP valve-controlled perfusion system equipped with a QMM-8 outlet manifold (ALA Scientific Instruments, Farmingdale, NY).

## Confocal microscopy

To examine proper membrane trafficking of mutant prestin mutants, live CHO cells transfected with GFP-fused prestin constructs were imaged with a Zeiss LSM710 confocal microscope with GFP excitation at 488 nm.

## Voltage clamp fluorometry

**Preparation and injection of Xenopus oocytes.** The surgery followed standard procedures and was carried out in accordance with the German law of animal protection and following the guidelines and regulations of the local authority of the state of Hesse (Regierungspräsidium Gießen, approval A16/2019). Oocytes were harvested from *Xenopus laevis* frogs of our own colony anesthetized in phosphate-buffered water containing 0.16% 3-aminobenzoate methanesulfonate salt.

Oocytes were injected with 50 nl RNA (0.5 μg/μl) and incubated at 12 to 18 °C for 3–4 days in ND96 medium containing (in mM): 96 NaCl, 2 KCl, 1.8 CaCl$_2$, 1 MgCl$_2$, 5 HEPES, 2.5 Na-pyruvate, and 100 mg/l gentamicin, adjusted to pH 7.5 with NaOH.

**Recordings of transport currents in oocytes.** Two-electrode voltage clamp recordings were performed with a TEC-10CX TEVC amplifier (npi electronic, Tamm, Germany) connected to a PC via an ITC-1600 data acquisition board (HEKA Elektronik, Lambrecht/Pfalz, Germany). Data acquisition was controlled with the WinWCP V5.2.7 software[73]. Data were sampled with 40 kHz and low-pass filtered with 10 kHz. Currents from untreated oocytes or oocytes labeled with MTS-TAMRA (see below) were recorded in response to voltage ramps (−110 to +60 mV; 0.5 s). During recordings, oocytes were superfused alternately with solutions containing 10 mM Cl$^-$ and either no sulfate (in mM: 10 NaCl, 86 Na-aspartate, 2 KCl, 1.8 CaCl$_2$, 1 MgCl$_2$, 5 HEPES; pH 7.5) or 10 mM sulfate (in mM: 10 Na$_2$SO$_4$, 10 NaCl, 86 Na-aspartate, 2 KCl, 1.8 CaCl$_2$, 1 MgCl$_2$, 5 HEPES; pH 7.5). Recording pipettes contained 3 M KCl.

**VCF recordings.** On the day of recording, oocytes were labeled at 4 °C for 30 min in a solution containing (in mM): 92 KCl, 0.75 CaCl$_2$, 1 MgCl$_2$, 10 HEPES, and 0.05 (2-((5(6)-Tetramethyl-rhodamine)carboxylamino) ethyl)methanethiosulfonate (MTS-TAMRA), adjusted to pH 7.5 with KOH. Following labeling, oocytes were washed three times in ND96 and stored at 12 °C until recording. Subsequently, a single oocyte was placed in a recording chamber with the dark pole facing the top for VCF recordings[48]. VCF recordings were performed with a BX51 upright microscope (Olympus, Tokyo, Japan) equipped with a water-immersion XLUMPlanFI objective (20X, NA 0.95, Olympus). TEVC recordings were performed as described above. A family of voltage steps (50 or 100 ms) ranging from 200 mV to −120 mV or from 200 mV to −250 mV with 10 mV increments was applied (holding potential, −60 mV). The surface of the oocyte was excited by green light from an LED (LED4E099, Thorlabs, Newton, NJ) during and ~500 ms before and after each voltage-step stimulation. Light was passed through a Cy3 ET filter cube (excitation 545/25 nm, dichroic 565 nm LP, emission 605/

70 nm; AHF Analysentechnik, Tübingen, Germany) and detected by a photo-diode (Thorlabs SM05PD2B). The current from the photodiode was amplified with an Axopatch 200B amplifier (Molecular Devices, Union City, CA) and filtered with 10 kHz. ND96 without gentamicin and Na-pyruvate was used as extracellular recording solution. In some recordings, extracellular superfusion of oocytes was transiently switched to solutions where 15 mM NaCl was substituted by 10 mM $Na_2SO_4$ (Fig. 5g–i) or 10 mM Na-salicylate was added (Fig. 6g–i).

## Data analysis

Experimental data were analyzed with Igor Pro (Wavemetrics, Portland, OR). Current and NLC traces shown in the figures represent averages from 2 to 10 subsequent individual recordings.

Fluorescence traces were normalized and bleaching and other voltage-independent decay components were fit with a double exponential function and eliminated from the traces. Fluorescence-voltage relationships (FVs) were obtained from the average fluorescence intensities of a 10 ms interval during each voltage step. For displaying purposes, fluorescence traces were smoothed by a boxcar filter (filter width, 1.5 ms).

Data are given as mean ± standard error (SEM). Statistical significance was assessed with Student's t-test for comparison between two individual conditions, and by using Dunnett's test for multiple comparisons with a single control. Significance is indicated as $*p \leq 0.05$; $**p \leq 0.01$; $***p \leq 0.005$.

## Reporting summary

Further information on research design is available in the Nature Portfolio Reporting Summary linked to this article.

## Data availability

Source data for Figs. 1b–d, 2a, b, 3b, 4f, and 6j, k are provided with this paper. Initial and final configurations for each simulation replicate are deposited as pdb files at https://jugit.fz-juelich.de/computational-neurophysiology/prestin_dynamics. All other source data are available from the corresponding authors upon request. Source data are provided with this paper.

## Code availability

In-house python scripts were used to analyze MD simulations; the history-labeled water accessibility analysis is provided at https://jugit.fz-juelich.de/computational-neurophysiology/prestin_dynamics. The In-house python scripts relied on numpy (1.24.2), scikit-learn (1.3.0), and MDAnalysis (2.5.0). ChimeraX (1.6) was used for protein visualization. IGOR Pro 8 was used to analyze patch-clamp and voltage-clamp fluorometry data, including simple custom-written routines. All code used will be made available by the corresponding authors upon request.

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

## Acknowledgements

This work was supported by the Deutsche Forschungsgemeinschaft (German Research Foundation) to J.-P.M. (MA 7525/2-1, as part of the Research Unit FOR 5046, project P2) and to D.O. (OL 240/4-2 and OL 240/8-1 as part of the Research Unit FOR 5046, project P3). M.F.K. was supported by a scholarship of the Deutscher Akademischer Austauschdienst (DAAD), the TOBITATE! Young Ambassador Program (MEXT, Japan) and Grant-in-Aid for JSPS Fellows (19J20169). We acknowledge the support of the Core Facility 'Protein Biochemistry and

Spectroscopy' of the Philipps University Marburg. The authors gratefully acknowledge the computing time granted through JARA on the supercomputer JURECA at Forschungszentrum Jülich and the supercomputer CLAIX at RWTH Aachen University.

## Author contributions

B.G.H. and P.L. performed computational studies. B.G.H., P.L., and J.-P.M. analyzed computational data. M.F.K., D.L.-S., J.H., B.-M.H., A.F., A.Q., T.K.B., and D.O. performed electrophysiological experiments. M.F.K., D.L.-S., J.H., T.K.B., and D.O. analyzed experimental data. J.-P.M., D.O., and B.G.H. wrote the manuscript. J.-P.M and D.O. jointly conceived the study and supervised the work.

## Funding

## Competing interests

The authors declare no competing interests.
