## [Peer Review File · Nature Communications]

REVIEWER COMMENTS

Reviewer #1 (Remarks to the Author):

This interesting manuscript explores the mechanism underlying the function of Prestin, the protein responsible for electromotility of outer hair cells of the inner ear. Despite advances made in recent Cryo-EM studies of members of the SLC26 protein family, important aspects of Prestin conformational cycle remain to be elucidated. Here, the authors employ cysteine accessibility mapping, cadmium cross-bridging analysis, and voltage clamp fluorometry to probe the nature of the conformational change separating a compact and an extended state identified by equilibrium MD simulations. The evidence pointing to an elevator-like translation of the transport domain relative to the scaffold domain is convincing. Demonstrating that this vertical translation is the cause of protein expansion within the membrane will require further studies. Nevertheless, the authors find evidence that the translation is involved in voltage sensing, an important step toward establishing the electromechanical coupling mechanism. Overall, the manuscript is well written, but there are several aspects of the text and figures that need clarification. The following points require particular attention:

- 1) The authors state that their equilibrium simulations reveal a wider conformational landscape than suggested by recent experimental structures of Prestin. However, comparisons between the CS and ES models and experimental structures are not provided, thus differences cannot be evaluated.
- 2) Information on statistical analysis is missing. What tests were performed to evaluate statistically significant differences (e.g., Fig 4H). When performing multiple comparisons (Fig. 4F), what type of post hoc correction was applied?
- 3) The PCA input appears to be inter-Calpha distances between every 5th residue in the TM-domain. This needs to be better explained. Are these the distances between residue n and residue $n + 4$ for n covering all residues of the TM-domain? What about distances between residues that might be close in 3D but are separated by more than 4 residues along the primary sequence?
- 4) The fact that the protein complex was embedded in a DOPC membrane should be mentioned in the Results section introducing the equilibrium simulations. Also, while it is appropriate to explain PCA in the Methods, Fig. 1B and Supl. Fig. 3 are difficult to understand without any explanation on what PC-1 and PC-2 are in the Results section.
- 5) In the Cd^{2+} mediated cross-bridging section, the authors state: "Although further positions in TM8 were explored systematically in combination with D485C, they either lacked appreciable NLC

(D485C/A345C, D485C/V353C), preventing detection of cross-bridge formation in this functional assay, or were insensitive to Cd²⁺ beyond the residual effect seen with D485C/M143C (D485C/I344C and D485C/S352C).” However, data for D485C mutants containing additional substitution A345C, V353C, I344C, or S352C do not appear in figures or supplemental material.

6) In Fig. 6G, the authors show that the F-V curve for rPres R510C-TAMRA is altered by the presence of salicylate. The author should add some discussion about why the F-V is reduced in size but not much shifted. One may expect that salicylate abrogates NLCs by inhibiting the ES to CS transition and that this should produce a significant shift in the F-V. Did the author exclude potential quenching effects of the fluorophore by salicylate?

7) It is unclear how the elevator movement described here can cause the change in membrane capacitance produced by SLC26 proteins like Prestin. While the area-motor mechanism involving changes in circumference of the protein complex can intuitively explain an expansion of the membrane surface, how does a conformational change orthogonal to the membrane result into a change in circumference in the membrane plane? If that aspect of the mechanism is not understood, that should be stated and caution should be taken when interpreting correlations between the up-down movement of residues in TM10/TM3 and NLC.

Minor points:

The color choice to represent MTS accessibility data in Fig. 2 makes it difficult to see bars and individual points for MTSET (light gray). The color choice in Suppl. Fig. 5 is much clearer.

The bottom panel of Fig. 2B looks incomplete. While the MTS accessibility of TM10 was published in a previous study, it would still be useful to show some numbers instead of an empty bar graph.

Fig. 4G is incompletely labeled. The identities of residues colored in yellow and light blue are not provided. Legend of Fig. 4H refers to “fittings the derivative of first-order Boltzmann function to NLC curves as shown in (D,E)” but no fitting curves appear in the figure.

Fig. 6C is expected to show the reorientation of R150 between ES and CS, but the residue is difficult to see in sphere representation of the same color as the backbone, especially for the ES.

In the Methods section the conc. of salicylate is stated to be 1 mM but in the corresponding figure legends it is 10 mM.

Reviewer #2 (Remarks to the Author):

Overview

The research investigates both outward and inward facing conformations and is distinguished by its originality. It is noteworthy for combining computational and experimental methods, with the computational work being presented first, which is a departure from the traditional approach. However, there is a major concern regarding the possible over- and misinterpretation of the observations. Based on the presented data, it appears that the computational work did not reveal any response to changes in membrane potential, which were absent from the simulations. The MD trajectories display a significant conformational variability, but this does not mean that the most extreme conformation describes a depolarized state in the absence of any membrane potential. These issues are further discussed in the "major concerns" section below.

If the authors have data capable of clarify concern 1 section "major concerns" below, the work will demonstrate a clear response to the absence/presence of chloride in the binding site, which is a positive finding. While the VCF experiments provided solid evidence of voltage-dependent conformational changes, these were restricted to the reorientation of the ARG150 residue in TM3.

Figures:

Remarks concerning Figure 1B:

1. The figure suggests that the counts, when transformed into the parameter δ , are expected to be very high in the vicinity of the CS and ES areas. Unfortunately, the orange and green representations obscure the populations. At a minimum, the authors should consider presenting the figure without these areas in the supplementary material. If the counts are low in the masked areas, an explanation should be provided.
2. The axes in the figure should indicate the power of PC-1 and PC-2, i.e., the amount of variability explained by each component. Additionally, the units of the axes should be clearly specified. GROMACS reports the projections of the eigenvectors in nm, whereas structural biologists typically use Å. Therefore, the units must be explicitly stated.

Major Issues:

In my opinion, the paper should not be published until the following two issues have been addressed.

1) Firstly, the abstract implies that the MD simulations examined transitions between the extended (ES) and compact (CS) states. However, there is limited evidence of such transitions in the paper. To support this claim, the authors should demonstrate the dynamics of the large conformational changes associated with these transitions, possibly through time series analyses. If this is not possible, the abstract should be revised to avoid overestimating the observations. It is worth noting that the existence of CS and ES states in itself is not a new finding, as they are roughly equivalent to the "Up" and "Down" states described in previous studies (refs: doi:10.1016/j.cell.2021.07.034 and doi:10.1038/s41586-021-04152-4), which are cited in the manuscript at lines 106-118, 259-260, and 274-275.

Based on the presented data, it appears that some conformations remained relatively close to the initial state, CS (PDB 7LGU), with PC-1 values ranging from 40-80. On the other hand, other conformations differed significantly, with PC-1 values ranging from -30 to -40. It is unclear if the latter ensemble corresponds to chloride-free constructs, and if so, why the authors did not mention it. Furthermore, there is no evidence to support the notion that the observed ES structure represents a native response of the apo protein (without chloride ions).

2) The conclusion of section 1 (lines 212-213) asserts that the findings support the idea that an elevator-like reorientation drives eM, and this has prompted the authors to explore the reorientation experimentally under conditions that include the presence and experimental control of the membrane potential. However, there is no direct connection between issue 1 (the absence/presence of chloride in the anion binding site) and the membrane potential. To investigate the effects of membrane potential, the authors could have subjected their simulation systems to various voltages, but this was not done. Therefore, to avoid overinterpretation, the paragraph and other sections that relate the comparisons between ES and CS to the membrane potential should be rephrased. It is important to note that the VCF experiments offer solid indications of voltage-dependent conformational changes, but these are limited to the reorientation of an Arg residue in TM3.

Method section:

Methods:

- Lines 812-813

“System equilibration was determined by monitoring the protein-lipid contacts until they converged” <-> it is not clear, which parameter was used to test the convergence. In other words, what the authors really monitored under the expression: “protein-lipid contacts”. Many things can be understood as contacts. The data supporting the convergence should be shown, at least in the supplm. material section.

- Salt concentration:

Why 200 mM? according to cochlea.eu nearby 150 would be more appropriate, especially when lines 131-132 state that physiological ionic concentrations were used.

- Replicate and statistics

The length of each of the (six?) replicated trajectories should be indicated. Did all the PCI ended up in the CS and all the PCIF in the ES? From the data presented, it is impossible to know.

- Line 799: please replace using “gmx membed” with using “gmx mdrun membed” and provide the required details of the used membed.dat table.

Minor remarks

L84: a missing parenthesis

L85-86 replace “in response membrane potential” by “in response to membrane potential”

L99 ☐ introduce the ‘TM’ abbreviation, transmembrane

L258: please replace “... sampling of these inward-facing states...” with “... sampling of inward-facing states...”, because there is no proof that the native conformation(s) correspond exactly to the cluster identified by the study.

Line 670: please replace ‘indicate’ with ‘indicates’

Line 693: Not D150C but R150C

First and foremost we would like to thank both reviewers for their time and expertise.

Reviewer #1

(Major points)

1) The authors state that their equilibrium simulations reveal a wider conformational landscape than suggested by recent experimental structures of Prestin. However, comparisons between the CS and ES models and experimental structures are not provided, thus differences cannot be evaluated.

We included a new supplemental figure (**Supp. Fig 2**) making extensive quantitative and visual structural comparisons between the CS-centroid and ES-centroid to experimental structures from Ge et al., Bavi et al., and Butan et al. This figure includes comparisons of the transmembrane domain (TMD) of CS-centroid, ES-centroid, and the experimental structures – RMSD of the TMD, as well as various structural quantities pertaining to the landscape (G82-S505 distance, TD-SD Z-shift, and angles between TM3-TM10, TM5-TM6, and TM6-TM14).

2) Information on statistical analysis is missing. What tests were performed to evaluate statistically significant differences (e.g., Fig 4H). When performing multiple comparisons (Fig. 4F), what type of post hoc correction was applied?

Student's t-test was used for evaluating significance between individual conditions in experiments (e.g. **Fig. 4H**). We used Dunnett's test for the multiple comparisons (e.g. **Fig 4F**, and **Figs.2A** and **3A**, where significance levels are now given explicitly). Information on statistical tests was added to the Methods section.

3) The PCA input appears to be inter-Calpha distances between every 5th residue in the TM-domain. This needs to be better explained. Are these the distances between residue n and residue n + 4 for n covering all residues of the TM-domain? What about distances between residues that might be close in 3D but are separated by more than 4 residues along the primary sequence?

Thank you for bringing this ambiguity to our attention, the text has been revised for clarity on **page 15, lines 660-664**. Our goal was to observe trends/correlations in global movements of the transmembrane domain (TMD), thus, inclusion of all residues wasn't necessary to extract global dynamics. Indeed, PCAs calculated from the entire TMD (excluding loops; 350 residues) and every 5th residue (excluding loops; 76 residues) yield equivalent results (data not shown). In either case (350 residues vs. 76 residues) all inter-C α distances are included in the input-features to PCA.

4) The fact that the protein complex was embedded in a DOPC membrane should be mentioned in the Results section introducing the equilibrium simulations. Also, while it is appropriate to

explain PCA in the Methods, Fig. 1B and Supl. Fig. 3 are difficult to understand without any explanation on what PC-1 and PC-2 are in the Results section.

The text has been revised for clarity. Mention of the embedding in a POPC membrane has been added to the results on **page 5, line 151**. The explanation for the meaning behind the principal components PC-1 and PC-2 has been added to the results on **page 5 lines, 155-159**. As ES and CS represent extremes of the largest correlated motions (i.e., first principal component), we can look at cluster-dependent structural observables to understand the complex underlying motions (**Fig 1 D, Supp. Fig 1 C-F, Supp. Fig 2 D-J, Supp. Fig 3 B, C**).

5) In the Cd²⁺ mediated cross-bridging section, the authors state: “Although further positions in TM8 were explored systematically in combination with D485C, they either lacked appreciable NLC (D485C/A345C, D485C/V353C), preventing detection of cross-bridge formation in this functional assay, or were insensitive to Cd²⁺ beyond the residual effect seen with D485C/M143C (D485C/I344C and D485C/S352C).” However, data for D485C mutants containing additional substitution A345C, V353C, I344C, or S352C do not appear in figures or supplemental material.

We performed additional experiments to determine any effects of mutants D485C/A345C, D485C/V353C, as well as I348C and succeeded in obtaining useful recordings despite the small signal amplitudes recorded with these mutants (variance is somewhat larger with some of the mutants, likely due to small signal-to noise). We found no significant effects of Cd²⁺ with these mutants. These data as well as those previously mentioned (D485C/I344C and D485C/S352C) are now given in revised **Suppl. Fig. 9F** and the main text has been revised accordingly:

‘Further positions in TM8 (I344C, A345C, I348C, S352C, and V353C; **Fig. 4G**) were explored systematically in combination with D485C/M143C. These cysteine pairs were insensitive to Cd²⁺ beyond the residual effect seen with D485C/M143C (**Supplementary. Fig. 9F**). Although the α -carbons of the latter cysteine pairs are predicted with a distance sufficiently small to form a Cys-Cd²⁺-Cys bridge (below 12 Å; ⁴¹⁻⁴³) the orientation of the side chains relative to the TM8/TM14 interface may explain the lack of Cd²⁺ cross-bridging.’

6) In Fig. 6G, the authors show that the F-V curve for rPres R510C-TAMRA is altered by the presence of salicylate. The author should add some discussion about why the F-V is reduced in size but not much shifted. One may expect that salicylate abrogates NLCs by inhibiting the ES to CS transition and that this should produce a significant shift in the F-V.

Ineed, the voltage dependence is shifted to more positive potentials by about 60 mV, which is consistent with previously published data of prestin’s voltage dependence measured either as NLC (e.g., Oliver et al., 2021, Homma & Dallos, 2011) or electromotility (e.g. Homma & Dallos, 2011). Due to the large voltage range displayed and the rather shallow voltage dependence, this may not easily be recognized in **Fig.6G**. We have therefore now added a new panel (**Fig. 6H**) to clearly show this shift and mention this important point in the main text.

'In accordance with previous measures of prestin's electromotile activity (i.e., NLC^{18,49} or electromotility⁴⁹), $V_{1/2}$ of the VCF dynamics was shifted to a more depolarized membrane voltage (**Fig. 6H**), consistent with salicylate acting by changing voltage-dependent distribution between compact and extended states⁴⁹.'

Did the author exclude potential quenching effects of the fluorophore by salicylate?

We thank the reviewer for drawing our attention to the possibility of fluorescence quenching by salicylate. We performed additional experiments to address this concern and found that indeed salicylate at the concentration used (10 mM) substantially reduced TAMRA fluorescence by about 6.5 % in TAMRA-labeled rPres-R150C-expressing oocytes (new **Suppl. Fig. 11A**). In order to check whether the effects of salicylate on prestin-VCF behavior might 'artificially' result from non-specific quenching rather than from actual meaningful protein-mediated effects (i.e. interaction between salicylate and the protein), we looked for a control condition. Therefore, we tested the effect of salicylate on the fluorescence intensity of a voltage-dependent and TAMRA-labeled protein that has not been reported to be sensitive to salicylate. To this end, we used the voltage sensitive phosphatase *ciVSP*(G214C) as a reference, which has been studied extensively by VCF (e.g., Kohout et al., 2008). The fluorescence reduction by salicylate in TAMRA-labeled *ciVSP*-expressing oocytes is similar to that of TAMRA-labeled prestin (new **Suppl. Fig. 11B**, left panel). However, the normalized fluorescence data show that salicylate has no impact on the voltage-induced fluorescence changes of *ciVSP* (new **Suppl. Fig. 11B**, right panel). This is in stark contrast to rPres.

In addition, salicylate has only a minor effect on the voltage dependence of the VCF signal in *ciVSP* (-10 mV; new **Suppl. Fig. 11C,D**). In contrast, salicylate has a large effect on the voltage dependence of the VCF signal in rPres (more than +40 mV, see answer above, **Fig. 11D**, and new panel **Fig. 6H**), indicating that changes of the (normalized) voltage-induced fluorescence signals do indeed result from the specific interaction of salicylate with prestin rather than from quenching TAMRA fluorescence.

We added a brief reference to these control measurements to the main text (**lines 436-441**).

7) It is unclear how the elevator movement described here can cause the change in membrane capacitance produced by SLC26 proteins like Prestin. While the area-motor mechanism involving changes in circumference of the protein complex can intuitively explain an expansion of the membrane surface, how does a conformational change orthogonal to the membrane result into a change in circumference in the membrane plane? If that aspect of the mechanism is not understood, that should be stated and caution should be taken when interpreting correlations between the up-down movement of residues in TM10/TM3 and NLC.

In addition to modification of the text, we provide a supplemental figure (see Reviewer 1, major point 1, **Suppl. Fig 2D**) which demonstrates a correlation between the expansion of the TMD in

the membrane plane and an orthogonal shift of the transport-domain (TD) relative to the scaffold-domain (SD). In **Figure 1D,E**, we show that there is a concerted vertical shift and rotation of the TD with respect to the SD (TD-SD). The TD-SD rotation results in the extrusion of TM1 (in TD) – the longest intraprotein distance in the membrane plane is roughly between G82 (TM1, i.e., TD) and S505 (TM14, i.e., SD). Given that SD is relatively stable, the TD-SD vertical shift (and therefore rotation) results in an expansion of the TMD in the membrane plane. We further demonstrate a cluster-dependent expansion of the membrane from CS to ES, measured as the excluded lipid area, i.e. the number of lipids ‘pushed’ away from the protein as a result of expansion (**Fig 1D right**).

However, it should be noted that the voltage-dependent capacitance (NLC) used here as the experimental readout for prestin’s dynamics does not result from the membrane-area change but from the transmembrane movement of ‘sensing charge’ that moves orthogonal to the membrane in response to changes of the membrane potential, which is - generally speaking - also in line with the proposed elevator mechanism. This origin of NLC is reflected in its bell-shape (derivative of Boltzmann function) rather than sigmoidal voltage dependence. Thus, while all MD simulations are performed under equilibrium conditions (i.e., 0 mV), NLC rather measures conformation-dependent charge movement of the TMD along the membrane normal in response to voltage perturbation. We now included an analysis of the protein-charge distribution of the transmembrane domain in our simulations; this analysis predicts an outward charge movement upon ES–CS transition, as expected from electrophysiological experiments (**Suppl. Fig 3C**).

(Minor points)

The color choice to represent MTS accessibility data in Fig. 2 makes it difficult to see bars and individual points for MTSET (light gray). The colors choice in Suppl. Fig. 5 is much clearer.

We chose grey color coding for MTSES/MTSET in this figure, because colors are already being used to highlight intra- versus extracellular accessibility (blue/red), and to signify ES versus CS states (green/yellow). Also, we like to mention that the distinction between MTSES and MTSET-induced effects is not of specific importance here, because changes in response to either reagent are equivalently taken as evidence for accessibility.

Although we considered alternative color-coding schemes, we did not find one that is easier to read and therefore prefer to stay with the initial scheme.

The bottom panel of Fig. 2B looks incomplete. While the MTS accessibility of TM10 was published in a previous study, it would still be useful to show some numbers instead of an empty bar graph.

The comment appears to refer to **Fig. 2A**, lower panel (rather than Fig. 2B). We have now added a complete dataset, which was not published in Gorbunov et al., 2014, where only example traces had been shown. Additionally, for one mutant (L401C), supplementary data on MTS-induced shifts of

voltage dependence was added to **Suppl. Fig. 2F**, because in this mutant MTS reactivity leads to a $V_{1/2}$ shift rather than reduction of NLC amplitude.

Concerning **Fig. 2B**, columns that appear empty indeed show values of extracellular water accessibility that are virtually zero within the simulated time-scales. We have now added the individual data points (violin plots) as requested by Nature Communications guidelines, which also improves recognition of these small values.

Fig. 4G is incompletely labeled. The identities of residues colored in yellow and light blue are not provided.

Thanks for spotting the missing description, we have now changed color coding to account for the additional data (response to point 5, see above), and added an appropriate description in the revised legend.

Legend of **Fig. 4H** refers to “fitting the derivative of first-order Boltzmann function to NLC curves as shown in (D,E)” but no fitting curves appear in the figure.

We added the fitting curves to **Fig. 4D,E**.

Fig. 6C is expected to show the reorientation of R150 between ES and CS, but the residue is difficult to see in sphere representation of the same color as the backbone, especially for the ES.

We changed the structural representation in **Fig. 6C** to a lateral view as well as coloration of residue R150, which we think facilitates recognition of movement of this position, specifically with respect to translocation normal to the membrane.

In the Methods section the conc. of salicylate is stated to be 1 mM but in the corresponding figure legends it is 10 mM.

Thanks for spotting this mistake. The concentration was in fact 10 mM, which is now corrected in the Methods section.

Reviewer #2

Figures:

Remarks concerning Figure 1B:

1. The figure suggests that the counts, when transformed into the parameter δ , are expected to be very high in the vicinity of the CS and ES areas. Unfortunately, the orange and green representations obscure the populations. At a minimum, the authors should consider presenting the figure without these areas in the supplementary material. If the counts are low in the masked areas, an explanation should be provided.

We utilized the density based spatial clustering of applications with noise (DBSCAN) algorithm, which necessarily identifies regions of relatively high density. The DBSCAN parameters used in this analysis are in the methods on **page 16, line 668**. A summary of the clusters from the DBSCAN output are provided in the methods on **page 16, lines 668-673**. The two clusters chosen for analysis, CS and ES, are the two largest clusters encompassing 17% and 25% of the total simulated data respectively. The 2D conformational landscape is provided in the absence of any overlaid clusters in a new supplemental figure (**Suppl. Fig 3A**).

2. The axes in the figure should indicate the power of PC-1 and PC-2, i.e., the amount of variability explained by each component. Additionally, the units of the axes should be clearly specified. GROMACS reports the projections of the eigenvectors in nm, whereas structural biologists typically use Å. Therefore, the units must be explicitly stated.

Principal components PC-1 and PC-2 account for 35% and 19% of the total variance respectively; this has been clarified in the results on **page 16, line 666-667**. Since the units in the underlying data (inter-C distances) are in Å, the projections will also carry Å – all figures with PCA landscapes have been modified accordingly.

Major Issues:

In my opinion, the paper should not be published until the following two issues have been addressed.

The text has been revised to reflect the following point-by-point changes. Additionally, a new supplemental figure is provided to help ameliorate the following issues. We thank the reviewer for raising these concerns, as we believe the additional figures and textual clarifications have greatly improved the manuscript.

1) Firstly, the abstract implies that the MD simulations examined transitions between the extended (ES) and compact (CS) states. However, there is limited evidence of such transitions in the paper. To support this claim, the authors should demonstrate the dynamics of the large conformational changes associated with these transitions, possibly through time series analyses. If this is not possible, the abstract should be revised to avoid overestimating the observations.

Our present aim was to sample the relevant conformational space of prestin using unbiased MD simulations in order to identify conformational changes that can explain electromotility. During all unbiased simulations of prestin, initialized in the compact state (PDB ID: 7LGU), we see a transition from the CS to an expanded state (i.e., PC-1 < 0), including but not limited to the ES. Though the reverse transition (ES–CS) is not observed in the simulated time-scale, it does not prevent structural comparisons between the CS and ES and relating those with experimental data. We have provided a new supplementary figure which provides a visualization of the trajectories in PCA space showing the CS to ES transition originating from both PCI and PCIF simulations (**Suppl. Fig 3D-I**).

In future studies, it will be important to expand these qualitative insights and to quantify the kinetics and thermodynamics of these conformational transitions using molecular simulations.

It is worth noting that the existence of CS and ES states in itself is not a new finding, as they are roughly equivalent to the "Up" and "Down" states described in previous studies (refs: doi:10.1016/j.cell.2021.07.034 and doi:10.1038/s41586-021-04152-4), which are cited in the manuscript at lines 106-118, 259-260, and 274-275.

The structures comprising the CS cluster from our simulations are most similar to the 'Up' and 'Intermediate' states found in Ge et al., Bavi et al., and Butan et al. (**Suppl. Fig 2A-G**), however, the ES cluster identified in our simulations has structural characteristics which are distinct and beyond the experimentally resolved 'Down' states – evidenced by an expansion in the membrane plane (G82-S505 distance) (**Fig. 1C,D(right), Suppl. Fig. 1C, Suppl. Fig 2D**), rotation of the transport domain relative to the scaffold domain (TD-SD angle) (**Fig. 1D(left),E, Suppl. Fig. 2A-C**), vertical movements (TD-SD z-shift) (**Fig. 1D(center),E, Suppl. Fig. 2D**), and a novel kink between TM3 and TM10 (**Suppl. Fig. 1E, Suppl. Fig. 2A-C,E**), further opening the anion binding pocket to the intracellular solution (**Suppl. Fig. 6Q**).

Based on the presented data, it appears that some conformations remained relatively close to the initial state, CS (PDB 7LGU), with PC-1 values ranging from 40-80. On the other hand, other conformations differed significantly, with PC-1 values ranging from -30 to -40. It is unclear if the latter ensemble corresponds to chloride-free constructs, and if so, why the authors did not mention it.

The Prestin_Cl-Free (PCIF) simulation was constructed by excluding the bound chloride from the already constructed Prestin_Cl-Bound (PCI) simulation (and further neutralized), which was built from PDBID: 7LGU (compact/UP state from Ge et al.). Thus, all simulations were initialized in the conformational landscape within the CS cluster. Analysis of the trajectories in the conformational landscape, show that all simulations eventually exit the CS cluster within the simulated time, regardless of the initial bound state of the ion. (**Suppl. Fig. 3D-I**).

Furthermore, there is no evidence to support the notion that the observed ES structure represents a native response of the apo protein (without chloride ions).

Results from our simulated water accessibility analysis (introduced on **page 6, line 227-239**), when decomposed into CS and ES are in good agreement with the MTS accessibility results (**Fig. 2A,B, Suppl. Fig. 6A-0**). Our cadmium cross-linking experiments, designed from the CS and ES RMS-centroids, are in excellent agreement with the distribution of inter-C α distances from the CS and ES clusters (**Fig. 4**). Here we provide two pieces of experimental evidence corroborating the ES structure.

Though the PCIF simulations leave the region encompassed by CS after a short amount of time (<100 ns), all PCI simulations also leave the CS ($40 \leq PC-1 \leq 80$, $-20 \leq PC-2 \leq 20$) region, towards either ES ($-55 \leq PC-1 \leq -15$, $-40 \leq PC-2 \leq 0$), or other regions of the conformational landscape which adopt an expanded state – where PC-1 is roughly less than 0 (**Suppl. Fig. 3D-I**).

Thus the observed ES structures, while initially only observed in PCIF simulations, encompass trajectory frames initialized in PCI and PCIF (**Suppl. Fig. 3D-I**).

To address the reviewers concern regarding chloride binding we developed a chloride-binding analysis (**page 17, lines 717-727**); both PCI and PCIF simulations experienced reversible binding/unbinding with varying rates when $PC-1 < 0$. We mapped the results of the chloride-binding analysis to the PCA landscape to visualize the conformation dependence on chloride binding (**Suppl. Fig. 3B**).

2) The conclusion of section 1 (lines 212-213) asserts that the findings support the idea that an elevator-like reorientation drives eM, and this has prompted the authors to explore the reorientation experimentally under conditions that include the presence and experimental control of the membrane potential. However, there is no direct connection between issue 1 (the absence/presence of chloride in the anion binding site) and the membrane potential.

We agree with the reviewer regarding the importance of understanding prestin's dependence on chloride for the eM cycle and future studies should address this; however, we believe it is ultimately outside of the scope of this study. In the present study we don't interrogate the kinetics/thermodynamics of the CS to ES transition, but link structural data from the respective clusters to experiments in order to provide a molecular explanation of electromotility.

To investigate the effects of membrane potential, the authors could have subjected their simulation systems to various voltages, but this was not done. Therefore, to avoid overinterpretation, the paragraph and other sections that relate the comparisons between ES and CS to the membrane potential should be rephrased.

Indeed, molecular simulations with applied transmembrane potentials could be a possible approach to study the molecular basis of electromotility. However, for reasons of efficiency, we decided to obtain similar insights by analyzing the transmembrane (intra-protein) charge movement across the membrane that is associated with conformational transitions and which is expected to link the free-energy changes between conformations to transmembrane voltages.

We now included a charge-transfer analysis, as a proxy for the "sensing charge" (**Suppl. Fig. 3C**). This analysis demonstrates that the transition from ES to CS states transition would produce an outward current and thus qualitatively explain the experimentally observed voltage dependence. Furthermore, these results provide corroborating evidence that the voltage-dependent change in fluorescence, which correlates with electromotility, is due to an elevator movement of the SD relative to the TD demonstrated by the CS and ES clusters found under equilibrium conditions. This analysis is now introduced on **page 5, lines 177-182**.

While no membrane potential was used in the simulations, we find conformation-dependent changes in water-accessibility (**Fig. 2B, Suppl. Fig. 6A-0**), area-expansion (**Fig 1D (right)**), and charge-movement (**Suppl. Fig. 3C**). While an applied voltage is expected to alter the relative free-energy of the landscape, it wouldn't alter the conformations accessible to the protein – as we are not making any quantitative statements about the thermodynamics or kinetics of the

transition, we find the conformational space explored to be sufficient for making hypotheses and connections to experimental data.

It is important to note that the VCF experiments offer solid indications of voltage-dependent conformational changes, but these are limited to the reorientation of an Arg residue in TM3.

Given that R150 is located at the (extracellular) periphery of the membrane, i.e. outside the membrane electrical field, its voltage-driven reorientation indicates movement of a larger structural unit or domain that involves transfer of electrically charged moieties across the membrane electrical field, i.e. with a substantial vectorial component normal to the membrane. Thus the VCF signal is unlikely to report on a local reorientation of R150.

We like to note the similarity to VCF measurements of classical voltage sensor domains (e.g., as now shown in **Suppl. Fig. 11** with Ci-VSP as a control), where the fluorophore is bound to a residue at the outer end of a TM helix.

Taking together

- (i) the above notion that R150 signal reports on a domain reorientation
 - (ii) the observation that the VCF signal recapitulates all hallmarks of electromotility
 - (iii) the experimental evidence of movement between transport and scaffold domains (SCAM, cross-linking),
 - (iv) the elevator movement observed in MD simulations
- leaves us with the elevator movement underlying electromotility as the most parsimonious and robust conclusion.

We had tried to convey this logic in section 2 of the discussion (*'Nature of the electromotile molecular transition'*), and have now revised parts of this paragraph for more conclusive reasoning.

Method section:

Methods:

- Lines 812-813

“System equilibration was determined by monitoring the protein-lipid contacts until they converged” <-> it is not clear, which parameter was used to test the convergence. In other words, what the authors really monitored under the expression: “protein-lipid contacts”. Many things can be understood as contacts. The data supporting the convergence should be shown, at least in the supplm. material section.

The text has been modified to further clarify the parameters used for monitoring convergence via protein-lipid contacts in the methods on **page 15, lines 642-652**. These data have been added as a supplemental figure (**Suppl. Fig. 4**). In short, protein-lipid contacts are defined as the number of protein atoms, either from the whole protein or just the dimer-interface, which come within 5 Å of any atom of the lipid head-group.

- Salt concentration:

Why 200 mM? according to cochlea.eu nearby 150 would be more appropriate, especially when lines 131-132 state that physiological ionic concentrations were used.

We would like to emphasize that our current simulation study was not designed to provide exact quantitative estimates of the effects of chloride concentrations on protein dynamics (which is also unlikely supported by current classical mechanics force fields). To facilitate sampling of spontaneous Cl-binding in our simulations, we therefore used slightly higher-than-physiological concentrations.

To our knowledge, the precise concentration of cytosolic chloride in OHCs has not been measured experimentally. However, we agree that 150 mM would be more likely to reflect the physiological concentration than the 200 mM used in our MD simulations. We thus removed the statement 'physiological concentration'. While the precise Cl concentration quantitatively affects the distribution between states (e.g., Oliver et al, 2001), the slight relative deviation from physiological concentrations will not impact the fundamental behavior probed here by MD simulations.

- Replicate and statistics

The length of each of the (six?) replicated trajectories should be indicated. Did all the PCI ended up in the CS and all the PCIF in the ES? From the data presented, it is impossible to know.

Each replicate simulation length ranges from 830 ns to 1.16 μ s, giving a combined 6.2 μ s of conglomerate simulated time. Individual simulation lengths are noted in **Suppl. Fig. 3D-I**, and the total simulation time is mentioned on **page 5, line 149**. Visualization of the trajectories in PC-space show that 1 of 3 PCI replicates ends in ES, and 2 of 3 PCIF replicates end in ES – furthermore, none of the simulations remained in CS for more than 400 ns (**Suppl. Fig. 3D-I**).

- Line 799: please replace using "gmx membed" with using "gmx mdrun membed" and provide the required details of the used membed.dat table.

We have modified the text and used the correct naming, and have provided the details of the embedding.dat file used on **page 15, lines 629-631**.

Minor remarks

L84: a missing parenthesis

Parenthesis added.

L85-86 replace "in response membrane potential" by ""in response to membrane potential"
Corrected.

L99 introduce the 'TM' abbreviation, transmembrane

Changed as suggested.

L258: please replace "... sampling of these inward-facing states..." with "... sampling of inward-facing states...", because there is no proof that the native conformation(s) correspond exactly to the cluster identified by the study.

Changed as suggested.

Line 670: please replace 'indicate' with 'indicates'

Corrected.

Line 693: Not D150C but R150C

Corrected.

REVIEWER COMMENTS

Reviewer #1 (Remarks to the Author):

The authors did a great job with the revisions. I have no additional concerns.

Reviewer #2 (Remarks to the Author):

I readily acknowledge that I have gained valuable insights from the manuscript, particularly in terms of data analysis and experimental methodologies.

I am now in concurrence with the majority of the points raised, including aspects such as the PCA analysis and the trajectories it produces. It is understandable that achieving absolute certainty, to the extent of 100%, regarding the exact correspondence of the ES state to the cellular reality might be a challenge. However, this critique can be directed at any model, regardless of whether it is derived from experimental observations or molecular dynamics simulations. In light of this consideration, the alignment of results with MTS and CS experiments could potentially provide a satisfactory level of validation.

Nonetheless, regrettably and with some reluctance, I find myself compelled to reiterate the principal concern that I initially highlighted in my original review. As previously articulated, my prior comments read: ... : “The conclusion of section 1 claims (lines 212-213): “Notably, the findings support the idea that an elevator-like reorientation drives eM, prompting us to explore the reorientation experimentally, under conditions that also include the presence and experimental control of the membrane potential”. If issue 1) is addressed, the work will become able to show a response dependent on the presence/absence of chloride in anion binding site. Unfortunately, there is no direct link here with membrane potential. To address a response to the membrane potential, the authors could have applied various voltages to their simulation systems, which they did not. To not overinterpret the results, this paragraph as well as all other sections where the comparisons ES versus CS are interpreted in terms of membrane potential should be rephrased. Whereas the VCF experiments provide solid hints regarding voltage dependent conformational changes, e.g., these hints are limited to the reorientation of an Arg residue in TM3.”

In response to my previous comment, the authors made an adjustment by introducing a single word in lines 212-213 to address the identified issue. However, it is disheartening to observe that other segments of the manuscript have not undergone necessary corrections.

To clarify, I acknowledge the authors' decision not to embark on an exploration of the protein's reaction to membrane potential due to the extensive sampling and time requirements involved. Nevertheless, it is imperative that the presentation of results and conclusions refrains from suggesting an investigation

of such driving forces. For instance, consider lines 125-126, where it is stated: "To resolve the voltage dependence of prestin's conformational landscape and thereby identify structural transitions underlying eM, we probed the dynamics...". While the authors may assert that their intent was limited to the voltage dependence as associated with VCF, it remains evident that most readers may overlook these subtle distinctions and mistakenly assume an inquiry into voltage dependence, which is not the case.

Regarding the analysis of charge transfer, the presented evidence appears persuasive, as it demonstrates a discernible displacement of charges corresponding to an outward current. This could lead to the inference that such a phenomenon aligns with voltage dependence. However, I believe there exists a certain bias in this line of reasoning. The depolarization of membrane potential inherently attracts negative charges from a transmembrane protein toward the intracellular leaflet, or in the case of hyperpolarization, it could attract basic residues, for instance. However, the converse is not necessarily true: the mere occurrence of a conformational change accompanied by the movement of charges toward the intracellular (or extracellular) leaflet does not unequivocally establish a causal link between this conformational alteration and fluctuations in membrane potential. To substantiate this assertion, I replicated the methodology outlined in lines 730-744, as introduced earlier in lines 176-182. However, for this purpose, I opted to reproduce the calculations on the K⁺ channel KcsA, a recognized example of a voltage-independent channel. Following the steps of superposition, ion removal, and sequence restriction to identical residues (344 residues) from the open and closed channels with PDB accession numbers 7MHR and 7MUB, and subsequently applying equations 1, 2, and 3 delineated in lines 734-738 of the manuscript, the outcome reveals a charge displacement along the z-direction of 3.95 Å. This instance highlights that even a voltage independent protein can exhibit a notable charge transfer perpendicular to the membrane, leading to an observable current. Nevertheless, this charge transfer does not elucidate any voltage dependence within a voltage-independent channel.

I wish to emphasize that the manuscript maintains a commendable level of quality and elucidates crucial mechanisms. My intention does not oppose the concept of the elevator movement. However, I reiterate my appeal to the authors to revise the references to membrane potential to ensure that readers are not led astray into believing that this factor was indeed a subject of investigation.

We would like to thank both reviewers for their time and effort in carefully evaluating the revised manuscript.

Response to Reviewer #2

Before addressing reviewer #2's concerns, we want to thank them for their careful, critical, and thoughtful response to our work. We feel that the final revisions we have made strengthen the manuscript as a whole, and specifically remediate any misleading ambiguous phrasing regarding the use of a transmembrane voltage.

Nonetheless, regrettably and with some reluctance, I find myself compelled to reiterate the principal concern that I initially highlighted in my original review. As previously articulated, my prior comments read: ... : "The conclusion of section 1 claims (lines 212-213): "Notably, the findings support the idea that an elevator-like reorientation drives eM, prompting us to explore the reorientation experimentally, under conditions that also include the presence and experimental control of the membrane potential". If issue 1) is addressed, the work will become able to show a response dependent on the presence/absence of chloride in anion binding site. Unfortunately, there is no direct link here with membrane potential. To address a response to the membrane potential, the authors could have applied various voltages to their simulation systems, which they did not. To not over interpret the results, this paragraph as well as other sections where the comparisons ES versus CS are interpreted in terms of membrane potential should be rephrased. Whereas the VCF experiments provide solid hints regarding voltage dependent conformational changes, e.g., these hints are limited to the reorientation of an Arg residue in TM3."

In response to my previous comment, the authors made an adjustment by introducing a single word in lines 212-213 to address the identified issue. However, it is disheartening to observe that other segments of the manuscript have not undergone necessary corrections.

To clarify, I acknowledge the authors' decision not to embark on an exploration of the protein's reaction to membrane potential due to the extensive sampling and time requirements involved. Nevertheless, it is imperative that the presentation of results and conclusions refrains from suggesting an investigation of such driving forces. For instance, consider lines 125-126, where it is stated: "To resolve the voltage dependence of prestin's conformational landscape and thereby identify structural transitions underlying eM, we probed the dynamics...". While the authors may assert that their intent was limited to the voltage dependence as associated with VCF, it remains evident that most readers may overlook these subtle distinctions and mistakenly assume an inquiry into voltage dependence, which is not the case.

We greatly appreciate your concerns, and we concede that while a change in charge distribution along the membrane-normal may indicate voltage dependence, it is possible that the charge movement occurs outside of the voltage-dependent electric field and thus does not contribute to voltage sensing.

We thus performed additional computational experiments to explicitly address voltage dependence, as already suggested in the original review.

The gating charge (Q_g) is a quantity that links transmembrane voltage to the free-energy difference between two conformational states. Thus we sought to estimate Q_g for the ES \rightarrow CS transition using constant-electric-field simulations to induce transmembrane potentials of +/- 200 mV as a means of evaluating prestin's response to voltage computationally (**page 11, lines 447-457**) (Fig. 6J,K). Here we estimate Q_g using the difference in mean charge displacement of the two states, known as the Q-Route (see Methods).

The gating-charge is a system-level quantity which makes a decomposition to individual protomers of a dimer non-trivial. We therefore took careful precautions to ensure a robust and accurate estimate of the gating charge of an individual protomer between the CS and ES (see methods **pages 18-19, lines 776-788**). After generating representative CS and ES structures, we ran constant electric field simulations (3 x 100 ns simulations / voltage / system) totalling an additional 1.8 μ s of simulations distributed over a range of transmembrane potentials (-200 mV, 0 mV, 200 mV) – positional restraints were applied to all protein heavy atoms, allowing for the evaluation of the gating charge between the two states explicitly. Details of the additional simulations can be found in the methods on **page 18-19, lines 770-812**.

Discussion of the gating charge in the context of experimental data can be found on **pages 13-14, lines 574-580**. As in Bavi et al., we excluded chlorides from the anion binding pocket for the gating charge estimates. We feel the exclusion is valid, as we focus our analyses on the structural features of prestin with respect to electromotility. Certainly the underlying mechanisms of chloride dependent behaviors are of interest and warrant future studies, however it is ultimately outside the scope of this current study.

In summary, we have provided a quantitative description of the voltage coupling between the ES and CS states, estimating a gating charge of $0.62 \pm 0.02 e$, which is in good agreement with the experimental values of 0.6-0.8 e (Fig. 6J,K). This data corroborates intuitions gained from equilibrium simulations and reinforces the interconnection between simulation and experiment, i.e., demonstrating that the ES \rightarrow CS transition is indeed voltage dependent. Furthermore, our simulations, and subsequent discovery of the ES state, underscore the need for extended sampling beyond structures elucidated from CryoEM and other experimental methods – compare ES_{Qg} ($Q_g = 0.62 e$) to the “Down-I” state (PDB: 7S9B, $Q_g = 0.4 e$) (Bavi et al. 2021, Nature).

Finally, we emphasize that while including the gating-charge/voltage simulations to corroborate the voltage dependence of the CS-ES transition, we nevertheless adjusted the wording as suggested to avoid any misunderstanding (page 4, lines 125-129; page 5, lines 177-184).

Changes are highlighted in red in the revised manuscript file.

REVIEWERS' COMMENTS

Reviewer #2 (Remarks to the Author):

I'd like to express my gratitude to the authors for their excellent work in addressing the concerns and, in my view, enhancing the scientific rigor of their article. I no longer have any criticisms or recommendations to offer.

Answer to the reviewer (#2):

We appreciate the careful evaluation of our manuscript and the constructive suggestions for improvement.

(Remarks to the Author):

I'd like to express my gratitude to the authors for their excellent work in addressing the concerns and, in my view, enhancing the scientific rigor of their article. I no longer have any criticisms or recommendations to offer.

We thank the reviewer for agreeing with our revision.